# Non-destructive defect detection in powder metallurgy automotive oil pump stators using acoustic signals and machine learning classification

Mohammad Hossein Hadizadeh Isfahani[1], Hamid Moeenfard[1], Abbas Rohani[2*]

1 School of Mechanical Engineering, Ferdowsi University of Mashhad, Mashhad, Iran, 2 Department of Biosystems Engineering, Faculty of Agriculture, Ferdowsi University of Mashhad, Mashhad, Iran

* arohani@um.ac.ir

## Abstract

Defects such as cracks and mass reduction frequently occur during the production of powder metallurgy (PM) automotive oil pump stators, making rigorous inspection essential for reliable operation. Conventional human visual inspection is threshold-based, simple, and cost-effective, but it is limited by the presence of microscopic cracks, internal defects, and declining detection speed. In contrast, machine learning (ML) can automatically identify complex, non-linear patterns in acoustic signals, enabling more accurate and rapid defect detection. In this study, acoustic signals were recorded from 40 intact and 62 defective PM components, including 26 cracked, 16 with tooth breakage, and 20 completely fractured samples. Distinctive features were extracted from these signals and used to train multiple ML classifiers, including support vector machine, k-nearest neighbors, multilayer perceptron, and radial basis function (RBF) networks. Comparative evaluation revealed that the RBF network outperformed the other models, achieving 100% accuracy in distinguishing defective from intact components. This approach demonstrates that combining acoustic signal analysis with ML not only surpasses conventional inspection methods in accuracy and speed but also provides a scalable and reliable solution for industrial defect detection in PM components.

## Introduction

PM is a well-established process for producing components such as gears, cutting tools, electrical contacts, clutch plates, and magnets [1]. Because of its flexibility and ability to produce parts with tailored mechanical and physical properties, PM is extensively used in the automotive industry [2]. With global vehicle production approaching 100 million units annually, the demand for PM components has risen to nearly one million tons per year [3]. Since defects such as cracks and material inconsistencies

**Data availability statement:** All data files related to this study are available at the following URL: 10.6084/m9.figshare.30172591.

**Funding:** The author(s) received no specific funding for this work.

**Competing interests:** The authors have declared that no competing interests exist.

**Abbreviation:** Acc, Accuracy; MFE, Multidomain Feature Extraction; AE, Acoustic Emission Testing; ML, Machine Learning; ANN, Artificial Neural Network; MLP, Multi-Layer Perceptron; AUC, Area Under the Curve; MT, Magnetic Testing; BFGS, Broyden–Fletcher–Goldfarb–Shanno; NB, Naive Bayes; BP-NN, Backpropagation Neural Network; NN, Neural Network; BR, Bayesian Regularization; NDT, Non-Destructive Testing; CAE, Convolutional Autoencoder; OSS, One-Step Secant; CC, Cracked Components; P, Peak Frequency; CFC, Completely Fractured Components; P1–P4, P extracted from the four subranges; CGB, Conjugate Gradient Powell–Beale; A, Peak Amplitude; CGF, Conjugate Gradient Fletcher–Reeves; PA1–PA4, PA extracted from the four subranges; CGP, Conjugate Gradient Polak–Ribiére; PF, Peak Frequency; CNN, Convolutional Neural Network; PM, Powder Metallurgy; CTB, Components with Chipping or Tooth Breakage; Prec, Precision C, Spectral Centroid; RBF, Radial Basis Function; C1–C4, Spectral Centroid extracted from subranges (1: 1–5 kHz, 2: 5–10 kHz, 3: 10–15 kHz, 4: 15–20 kHz); Rec, Recall; EMD, Empirical Mode Decomposition; RF, Random Forest; F1, F1-Score; RP, Resilient Backpropagation; FFT, Fast Fourier Transform; S, Amplitude Skewness; FN, False Negatives; S1–S4, S extracted from the four subranges; FP, False Positives; SA, Skewness of Amplitude; GA-KELM, Genetic Algorithm–Improved Kernel Extreme Learning Machine; SC, Spectral Centroid; GD, Gradient Descent; SCG, Scaled Conjugate Gradient; GDA, Gradient Descent with Adaptive Learning Rate; SMO, Sequential Minimal Optimization; GDM, Gradient Descent with Momentum; Spec, Specificity; GDX, Gradient Descent with Variable Learning Rate; SVD, Singular Value Decomposition; GoogleNet, GoogleNet Architecture; SVM, Support Vector Machine; ICDW-YOLO, Improved Crack Detection in

can weaken strength, diminish performance, and increase the risk of failure, ensuring the structural integrity of PM components is critical [4]. Consequently, rapid and reliable non-destructive inspection techniques are essential for real-time quality control in large-scale manufacturing.

Robust quality assessment is essential in PM, where several non-destructive testing (NDT) methods have been investigated. Electrical resistance measurement provides rapid detection of surface and subsurface flaws in green-state components, with recent systems achieving sensitivities up to 7.6 mm deep [5]. Acoustic resonance analysis has emerged as a cost-effective and automated alternative to conventional techniques such as magnetic particle inspection, liquid penetrant testing, and X-ray imaging, offering fast, non-contact inspection suitable for production lines [4]. Ultrasonic testing also enables accurate subsurface defect detection and density mapping, with potential for robotic integration in large-scale manufacturing [6]. Thermal wave imaging has shown promise in detecting cracks in automotive transmission parts, particularly when combined with automated defect recognition algorithms, enabling high-resolution inspections at production speed [7]. Laser vibrometry offers location-independent crack detection through vibration spectra, although its industrial applicability and sensitivity to complex geometries remain under evaluation [8]. Although thermal wave imaging, ultrasonic testing, laser vibrometry, and resistance measurement offer specific advantages, acoustic resonance analysis is preferred for large-scale powder metallurgy production because it is automated, integrates seamlessly, and provides rapid and reliable detection of internal defects [4]. Its high speed, simplicity, and compatibility with robotic systems make it the most practical approach for quality control in high-volume powder metallurgy manufacturing.

Vibration analysis is a widely used non-destructive method for detecting cracks and mechanical defects by monitoring dynamic responses, including natural frequencies, amplitudes, and mode shapes. When combined with ML predictive maintenance [9]. In structural health monitoring, modal parameters that describe characteristic vibrational modes can be used to assess structural condition. For instance, changes in modal curvature successfully detected damage in prestressed concrete bridges, as validated on the Z24 bridge under different scenarios of the Brite-Euram project [10]. Similarly, continuous structural health monitoring using whirling modes and modal curvature analysis effectively detects stiffness degradation in rotary machinery and wind turbine blades, as confirmed by both simulations and experiments [11]. Among vibration-based methods, acoustic resonance testing evaluates a component's frequency response, which is highly sensitive to structural integrity. Cracks or discontinuities change natural vibrations, and extracting features from these signals enables reliable detection of internal defects. Historically, such changes were sometimes even audible, as in cracked bells producing altered pitches [12]. Advances in sensor technology have modernized acoustic resonance testing. Acoustic modal analysis can assess dynamic characteristics without altering intrinsic properties such as mass, stiffness, or damping [13]. This approach is particularly useful when installing accelerometers is impractical or would add mass that affects vibrations [14]. Acoustic resonance testing has also been used to assess nodularity in ductile iron

castings, offering full volumetric evaluation and higher detection accuracy compared with localized ultrasonic testing [15]. Although vibration-based methods are effective, some failure modes may be overlooked. Incorporating ML and neural networks allows for deeper analysis of complex data, reveals hidden patterns, improves diagnostic accuracy, and automates decision-making, thereby supporting efficient and cost-effective condition monitoring in industrial applications.

Wooden Materials using YOLO; TN, True Negatives; IRT, Infrared Thermography; TP, True Positives; ISDA, Iterative Single Data Algorithm; UT, Ultrasonic Testing; KNN, K-Nearest Neighbors; VAE, Vibration and Acoustic Emission Testing; L1QP, L1-Regularized Quadratic Programming; VBI, Vision-Based Inspection; LDA, Linear Discriminant Analysis; VBI+UT, Hybrid Vision and Ultrasonic Testing; LM, Levenberg–Marquardt; WPD, Wavelet Packet Decomposition; MA, Mean Amplitude; YI, Youden Index; MA1–MA4, MA extracted from subranges; YOLOv5, You Only Look Once version 5.

Recently, ML and artificial neural networks have gained attention in non-destructive testing. Combining acoustic resonance testing with time-frequency analysis and ML has enabled high-accuracy defect detection in sintered gears, even with limited data [16]. In weld inspection, integrating artificial neural networks with image processing and genetic algorithms improved spot weld strength assessment and fatigue evaluation, reducing the need for destructive testing [17]. Convolutional recurrent neural networks outperformed conventional ultrasonic testing in detecting complex internal defects in IN718 superalloy [18]. ML-enhanced laser ultrasonic testing, using principal component analysis and extreme gradient boosting, achieved a 98.48% detection rate for subsurface flaws [19]. Similar results were reported in adhesive bonding, where support vector machine models exceeded 90% accuracy [20], and in composite inspection, where autoencoders combined with unsupervised clustering analyzed over 2000 B-scans in under 1.3 seconds [21]. For powder metallurgy components, artificial neural network-based thermography achieved 85% accuracy for composite coatings [22] and hybrid convolutional neural network–support vector machine models improved surface defect detection in stators [23]. Deep learning techniques, including convolutional neural networks, autoencoders, and generative adversarial networks, are increasingly applied, with transfer learning and self-supervised approaches further enhancing performance [24]. In tapered roller powder metallurgy stators, radial basis function neural networks and adaptive neuro-fuzzy inference systems achieved 90.9% accuracy [25]. Multitask deep learning models effectively classified bearing faults under varying operating conditions [26]. Notably, zero-shot learning improved scratch classification accuracy from 60% to 98%, demonstrating substantial gains in diagnostic capability [27]. These advances highlight the transformative impact of artificial intelligence on defect detection, reliability, and predictive maintenance in industrial systems.

Although non-destructive testing is widely used in industrial quality control, the application of acoustic resonance analysis for defect detection in powder metallurgy and automotive components has been limited. Traditional methods, including visual inspection and image-based techniques, often involve high costs, depend on surface conditions, and are less effective in identifying internal flaws. Acoustic testing offers a faster and more practical alternative. When combined with ML, it enables automated feature extraction, reliable classification, and improved detection of subtle or hidden defects. This integration supports real-time, low-cost, and accurate inspection in large-scale manufacturing.

To address this need, the present study proposes a two-stage framework that combines acoustic resonance testing with ML. The method uses a wide set of frequency-domain features, including higher-order descriptors, and applies sub-band analysis to capture local

resonant modes. The framework performs binary classification to separate intact from defective components, followed by multi-class classification to identify cracks, tooth breakage, and fractures. The approach was validated on 102 samples (40 intact and 62 defective) using four classifiers: support vector machine, k-nearest neighbors, multilayer perceptron, and radial basis function networks. The results confirm that ML can substantially improve the accuracy and reliability of acoustic-based defect detection

## Materials and methods

Fig 1 presents the main steps of the proposed methodology for defect detection in PM automotive oil pump stators. The process began with sample preparation, where components were classified as either intact or defective. Defective samples were further grouped into three categories: cracked, chipped or broken teeth, and completely fractured. Each sample was then excited to generate acoustic vibrations using impact or drop excitation, and the signals were captured with a high-bandwidth microphone and converted into digital form. The recorded signals were filtered to remove noise and transformed into the frequency domain using Fast Fourier Transform (FFT). The spectrum was divided into four ranges (1–5 kHz, 5–10 kHz, 10–15 kHz, and 15–20 kHz), and five spectral features were extracted from each: mean amplitude, peak amplitude, peak frequency, amplitude skewness, and spectral centroid. This yielded 20 features per sample, which were normalized prior to analysis. The dataset was then divided into training (80%) and testing (20%) subsets. Four ML classifiers— support vector machine (SVM), k-nearest neighbors (KNN), multilayer perceptron (MLP), and RBF—were trained to perform both binary classification (intact vs. defective) and multi-class classification (cracks, broken teeth, fractures). Their performance was assessed using standard evaluation metrics, including confusion matrix, accuracy, precision, recall, F1-score, specificity, area under the curve (AUC), and Youden's Index. A comparative analysis was then conducted to determine the most accurate and reliable model. The following subsections provide detailed descriptions of each step.

### 2.1. Theoretical basis of natural frequency in defect detection

The intrinsic properties of components can indicate their structural health. Key parameters—including stiffness, damping, natural frequency, and mode shapes—have been widely used in defect detection studies for mechanical systems [28].

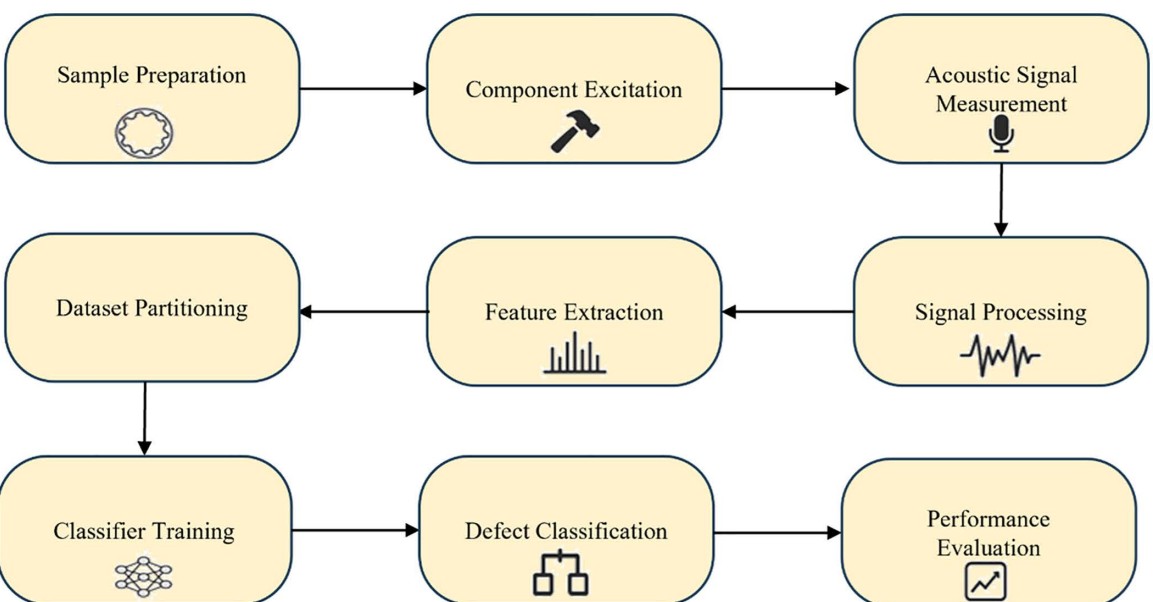

**Fig 1. Schematic flowchart of the proposed methodology for non-destructive defect detection in powder metallurgy automotive oil pump stators using acoustic signals and machine learning.**

Among these, natural frequency is particularly valuable for detecting defects. A defect can alter intrinsic properties such as natural frequency, mass, stiffness, or damping. A component can be modeled as a simple mass-spring-damper system, whose free vibration is described by [28]:

$$m\ddot{x} + b\dot{x} + kx = 0 \tag{1}$$

where $m$ is mass, $b$ is damping, and $k$ is effective stiffness.

The system's natural frequency is given by $\omega_n = \sqrt{(k/m)}$. Thus, natural frequency increases with mass reduction and decreases with stiffness reduction, such as that caused by cracks. By monitoring natural frequencies through modal testing, which also automatically extracts damping ratios, defects can be detected without separate measurements of mass or stiffness [4]. Modal testing is simpler and faster than direct stiffness measurement, explaining its widespread use in mechanical defect detection.

## 2.2. Acoustic modal analysis for defect detection

Experimental modal analysis is commonly used to measure a system's vibrational response, typically with accelerometers. However, in high-speed inspection environments, attaching and removing accelerometers can be time-consuming and reduce throughput. Components are often enclosed, restricting internal access and making accurate measurements difficult, and complex geometries further complicate sensor installation. Even for simple geometries, the accelerometer mass can be significant relative to the component, affecting measurement accuracy. Therefore, for defect detection in powder metallurgy components—which are low in mass and require rapid inspection—using accelerometers is impractical.

When a component is excited by an impact, its response can be expressed as a sum of mode shapes, each linked to a natural frequency. Thus, the system vibrates at its natural frequencies, and these vibrations transfer to the surrounding air, causing it to oscillate at similar frequencies. A microphone, which measures sound pressure level, can then be used to identify these natural frequencies. Experiments have shown that frequencies measured with a microphone are very close to those obtained with conventional accelerometers [16]. Acoustic modal analysis, also called acoustic resonance testing, provides clear advantages over accelerometer-based methods. In this approach, the component is excited with a broadband impact, and the microphone records the sound pressure level. Frequency analysis of the response yields the system's natural frequencies, which act as a fingerprint of the component [13]. Defects change these frequencies, so monitoring them allows evaluation of component health. Two main challenges are the limited placement of the microphone, which can reduce the signal-to-noise ratio, and the small delay in sound propagation from the impact point to the microphone. However, tests have shown that these issues have little effect on the accuracy of natural frequency and damping measurement [13]. Using a microphone avoids the problems of accelerometers, making it well-suited for the applications in this study.

## 2.3. Acoustic resonance testing procedure

The acoustic resonance testing procedure consists of the following steps:

**2.3.1. *Preparation of reference samples*.** In this study, ML methods are applied to evaluate component conditions. To train these methods, reference samples are prepared for classification. Two points are critical. First, intact and defective samples must be classified correctly, since errors at this stage can affect training and lead to inaccurate predictions. Second, using a wider range of defective samples improves training, as it helps the model generalize and recognize new types of defects. The components are divided into two main groups: intact and defective. The defective group is further divided into three subgroups: (i) cracked components, (ii) components with chipping or broken teeth, and (iii) completely fractured components. All reference samples are selected through visual inspection and penetrant liquid testing to confirm their condition.

                                                                          

**2.3.2. *Excitation of the component*.** The second step is to excite the component and record its acoustic vibration signals. The excitation must cover a wide enough bandwidth to stimulate all natural frequencies of the component. Two methods are commonly used: impact excitation and drop excitation. The choice between them depends on the component geometry and industrial constraints. For asymmetric components, repeatable testing requires precise alignment before impact excitation. Otherwise, changes in the impact location may cause inconsistent results. In large-scale production, however, this alignment is costly and impractical, so drop excitation is preferred. For symmetrical components, both methods are effective, although impact excitation is often favored because it can be easily automated. In this study, the impact method is applied for exciting powder metallurgy components

**2.3.3. *Measurement of acoustic response*.** The acoustic response of the component is recorded using a microphone. The signal is then stored in digital form for further analysis. Microphones with wide bandwidth, high signal-to-noise ratio, and adequate sampling rates (typically 20 Hz to 20 kHz) are preferred for accurate data acquisition.

**2.3.4. *Signal processing and frequency domain transformation*.** The recorded signals are in the time domain, where extracting useful features is difficult. To address this, FFT is applied to convert the signals into the frequency domain, which simplifies defect classification. Before the transformation, environmental noise is removed using a high-pass filter. This step enables the isolation of harmonic components in the signal, whose frequencies and amplitudes depend on the condition of the component. Monitoring these harmonics provides a basis for defect identification. In this study, defect detection is carried out using frequency-domain signals to achieve reliable classification of defective components.

**2.3.5. *Defect detection and feature extraction*.** Resonance frequencies and amplitudes are extracted from the frequency-domain signals to identify defects. In intact components, small deviations in resonance frequencies can occur due to manufacturing tolerances and minor geometric variations. In defective components, however, these frequencies change more noticeably depending on the type and severity of the defect. Analyzing these variations provides a basis for health assessment. Because natural frequencies alone are not always sufficient for classification in complex structures, additional spectral features are extracted. These features include:

1. *Mean amplitude (MA):* The average amplitude within the analyzed frequency range, indicating signal energy.

2. *Peak amplitude (PA):* The maximum amplitude in the frequency range, which may correspond to a resonance or defect.

3. *Peak frequency (PF):* The frequency at which the maximum amplitude occurs, possibly linked to natural frequencies or defect-related resonances.

4. *Skewness of amplitude (SA):* The degree of asymmetry in the amplitude distribution, which may reflect structural irregularities.

5. *Spectral centroid (SC):* The weighted center of the spectral distribution, showing how energy is spread across frequencies.

To support ML classification, the 20 Hz–20 kHz range is divided into four subranges: 1,000–5,000 Hz, 5,000–10,000 Hz, 10,000–15,000 Hz, and 15,000–20,000 Hz. From each subrange, the five features (MA, PA, PF, SA, SC) are extracted, giving 20 features in total. These are denoted as Peak Frequency (P1–P4), Peak Amplitude (PA1–PA4), Spectral Centroid (C1–C4), Mean Amplitude (MA1–MA4), and Skewness (S1–S4). The 20–1,000 Hz range is excluded to minimize the effect of environmental noise. Using these features improves the classification of intact and defective components, providing more reliable assessments of structural integrity. In this study, these features are applied directly for defect classification.

## 2.4. *Experimental setup and data acquisition*

To design an intelligent defect detection approach for PM components through acoustic resonance testing, the stator of an automotive oil pump from a Tiba vehicle was chosen as the case study. This component, manufactured in 2000 by the

Powder Metallurgy Factory for Saipa Automotive Company, has an average mass of 94 g and an outer diameter of 7 cm. Fig 2 presents three representative samples: (a) a cracked stator, (b) a stator with a missing tooth and reduced mass, and (c) an intact stator. The reliability of oil pump stators is critical because of their central role in the engine lubrication system. Defects in this component can cause pump malfunction, inadequate lubrication, accelerated wear of moving parts, overheating, and in severe cases, complete engine failure. Therefore, thorough inspection before installation is essential to ensure safe and reliable operation. In this study, defective and cracked stators were systematically collected from the production line over a six-month period. Careful inspection revealed that many of these samples displayed recurring defect patterns, particularly cracks along the internal surfaces of the stators. The repeated appearance of such flaws indicates that they are not random but are closely linked to weaknesses or inconsistencies in the PM manufacturing process. Specifically, variations in compaction pressure, sintering temperature, and cooling rate are suspected to be key contributors to defect formation. These findings highlight the importance of closely monitoring and optimizing these parameters to reduce variability and improve the overall reliability of PM components.

**2.4.1. *Equipment and instrumentation.*** To evaluate the condition of the selected components using acoustic resonance testing, the following equipment was used:

- *Microphone:* A high-bandwidth microphone is required to capture the resonance frequencies of the component. In this study, an ECM 8000 microphone (Behringer) with an omni-directional response, a frequency range of 15 Hz–20 kHz, and a sensitivity of –60 dB was used.

- *Sound card:* The analog acoustic signals must be converted into digital form for processing. An AudioBox Go sound card (Presonus) with a 24-bit/96 kHz sampling rate and USB-C connectivity was employed.

- *Computer:* A computer is needed for data analysis. MATLAB software was used in this study for signal processing and analysis.

- *Excitation mechanism:* To generate an acoustic response, the test samples required controlled excitation. Two methods were evaluated: dropping the sample from a fixed height onto a hard surface and applying an external impact force. The latter was selected for its higher repeatability, and a solenoid actuator (SMT-7074) was used for this purpose. The actuator operated at 90 W and 24 V, with a duty cycle of 25% and a stroke length of 11 mm, and its steel impact rod effectively excited the natural frequencies of the samples. During testing, each part was positioned 1 cm from the actuator tip, and a constant force of 10 N was applied to induce vibrations. Owing to the geometric symmetry of the outer surfaces, the

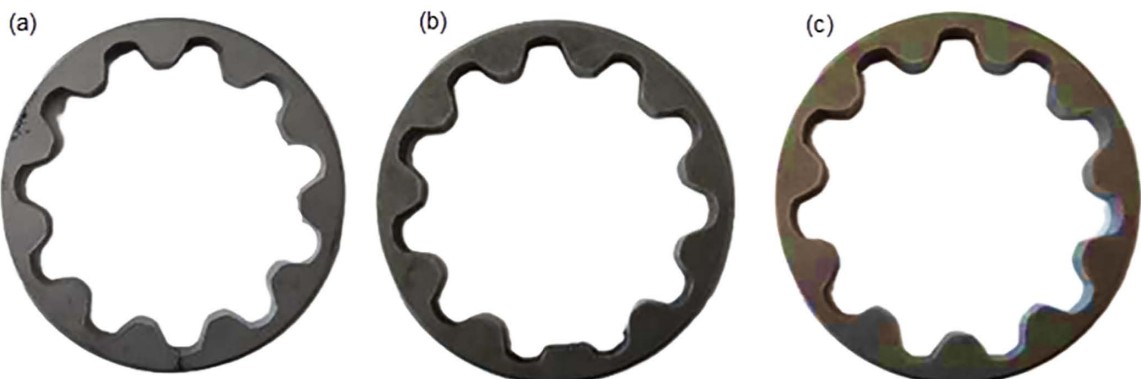

**Fig 2. Examples of oil pump stators manufactured using powder metallurgy, including (a) a cracked stator, (b) a mass-reduced stator, and (c) an intact stator.** Scale bar represents 5 cm. The outer diameter of each stator is 5.00 ± 0.05 cm.

angular position of the applied force had a negligible effect and was therefore selected randomly. The resulting vibrations generated acoustic waves, which were recorded using high-precision measurement equipment to ensure consistent signal acquisition and reliable detection of patterns associated with potential defects, thereby supporting reproducible testing conditions.

**2.4.2. *Experimental procedure*.** The laboratory setup for acoustic resonance testing is shown in Fig 3. The test sample was placed in front of the solenoid actuator. By pressing the green button, the solenoid was activated, and the steel rod struck the sample to produce an acoustic response. The generated sound was recorded by the microphone and transferred through the sound card to the computer for analysis.

**2.4.3. *Frequency domain analysis*.** As mentioned earlier, the condition of the test samples directly affects their frequency spectra. Before developing the classification algorithm, a preliminary validation was performed to confirm this effect. A representative time-domain signal from an intact sample is shown in Fig 4. It is difficult to extract defect-related information directly from the time-domain waveform. Therefore, a FFT was applied to convert the signal into the frequency domain. The resulting spectrum, limited to the microphone's operating range, is also shown in Fig 4. This spectrum contains three dominant frequency peaks with high amplitude and several smaller peaks.

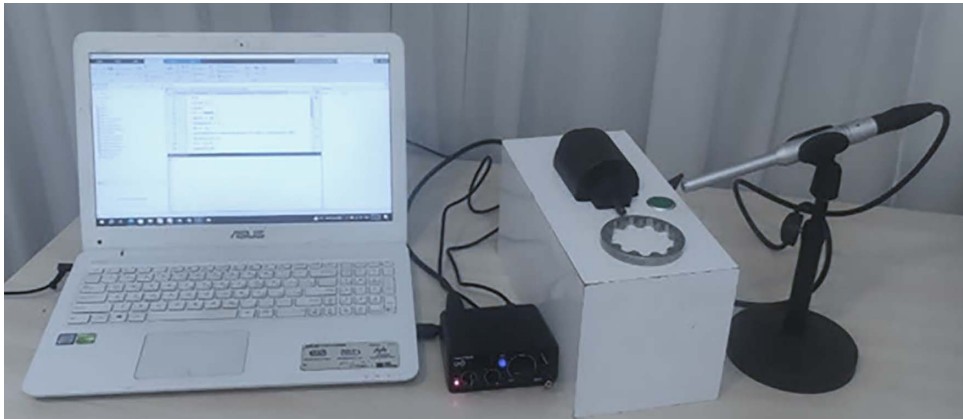

**Fig 3. Laboratory setup for acoustic resonance testing, showing the position of the test sample and the data acquisition process.**

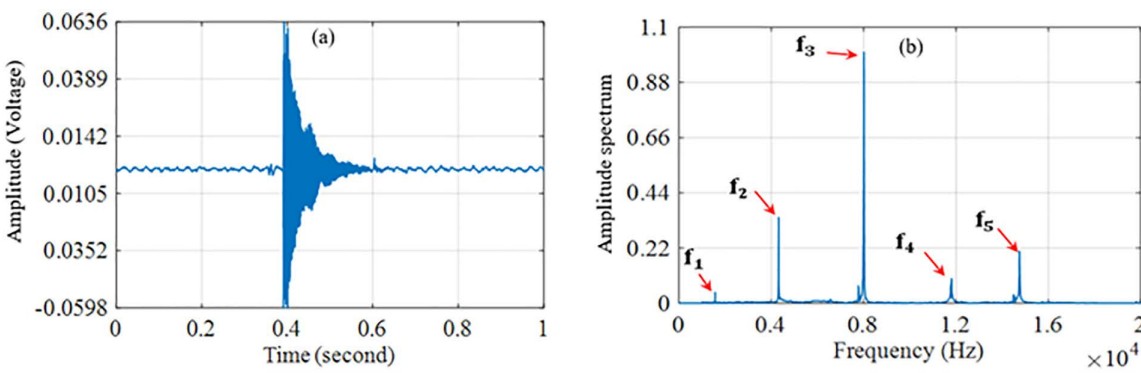

**Fig 4. (a) Frequency response (Hz) and (b) Time domain signal (s) of the microphone output after impact on an intact test sample.**

The five main frequencies, denoted as f1, f2, f3, f4, and f5, are identified as the natural frequencies of the sample with the highest amplitudes. The region around f3, which shows the strongest amplitude, is plotted in Fig 5 for intact, cracked, and mass-reduced samples. As expected, the peak frequency of the cracked sample is lower than that of the intact sample because of reduced stiffness. In contrast, the peak frequency of the mass-reduced sample is higher due to its lower mass.

## 2.5. *Implementation and performance evaluation of classifiers*

In this study, four classification models—Support Vector Machine (SVM), K-Nearest Neighbors (KNN), Multi-Layer Perceptron (MLP), and RBF neural network—were implemented to differentiate between intact and defective PM stators. The input space was constructed from four frequency ranges, within which five spectral features were extracted: mean amplitude, peak amplitude, peak frequency, skewness, and spectral centroid. This process yielded a total of 20 input features. To enhance comparability across features and improve classifier performance, all feature values were normalized to the range of [−1, 1]. The dataset was then randomly partitioned into training (80%) and testing (20%) subsets to facilitate classifier training and subsequent performance evaluation.

### 2.5.1. *K-nearest neighbor classifier*.
The K-nearest neighbor classifier assigns a test sample to the majority class of its K nearest neighbors in the feature space. The classifier operates in two main phases:

*Training Phase:* Each training sample is represented as an N-dimensional feature vector, and the corresponding class labels are stored. No explicit model fitting is required, as KNN is an instance-based, non-parametric method.

*Testing Phase:* Classification is performed in four steps:

1. The test sample is represented in the same N-dimensional space.

2. The K nearest neighbors are identified using a chosen distance metric.

3. The class labels of the neighbors are tallied.

4. The test sample is assigned to the majority class among its K neighbors.

The performance of KNN depends strongly on the distance metric, the number of neighbors (K), feature weighting, and data standardization. In this study, multiple distance metrics were evaluated, including Manhattan, Minkowski, Euclidean, Chebyshev, Standardized Euclidean, Correlation, Cosine, Hamming, Jaccard, and Mahalanobis. Feature weighting was

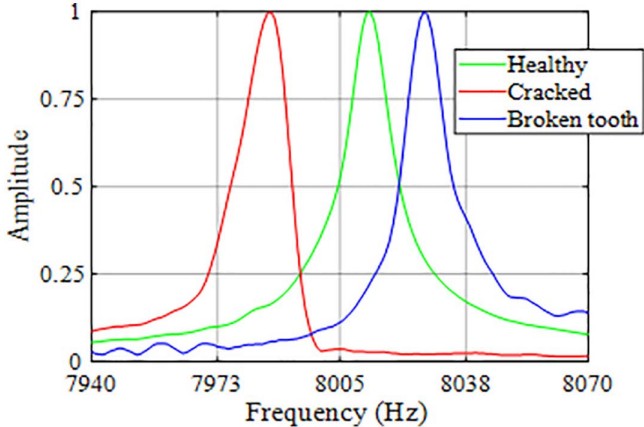

**Fig 5. Comparison of peak frequencies among intact, cracked, and mass-reduced samples.**

applied using the inverse squared distance method to emphasize more significant features. All features were standardized to ensure uniform scaling across dimensions.

**2.5.2. *Support vector machine classifier*.** The SVM classifier is well-suited for high-dimensional datasets. It works by identifying an optimal hyperplane that maximizes the margin between classes, thereby reducing misclassification. In this study, the SVM was employed for both binary classification (intact versus defective) and multi-class classification, distinguishing cracks, chipped or broken teeth, and fractures in PM stators.

To handle nonlinear relationships, four kernel functions were evaluated: linear, polynomial of degree 2, polynomial of degree 3, and the RBF. These kernels allow the input features to be mapped into higher-dimensional spaces, improving class separability. Their mathematical formulations are presented in Eq. (2)–(5):

$$K(x_i, x_j) = x_i^T x_j \tag{2}$$

$$K(x_i, x_j) = (\gamma x_i^T x_j + r)^2 \tag{3}$$

$$K(x_i, x_j) = (\gamma x_i^T x_j + r)^3 \tag{4}$$

$$K(x_i, x_j) = \exp(-\gamma \|x_i - x_j\|^2) \tag{5}$$

where $x_i$ and $x_j$ represent input feature vectors, $\gamma > 0$ denotes the kernel scale parameter that controls the influence of individual samples, and rrr is the coefficient used in the polynomial kernels. These kernel functions enable the SVM classifier to construct both linear and highly nonlinear decision boundaries, thereby improving classification performance in high-dimensional datasets.

During training, each sample was represented as an N-dimensional feature vector and labeled according to its class. The SVM model then determined the hyperplane that maximized the margin between the classes. Model parameters were iteratively optimized based on the selected kernel and solver.

Hyperparameter tuning was conducted systematically to enhance classification performance. The box constraint (C) was set to 1 for all models, controlling the trade-off between margin maximization and classification error. The kernel scale, $\gamma$ for RBF and polynomial offset, was set to 1 to control the feature space mapping, while the polynomial kernel offset was set to 0.1 to adjust the intercept. Solver algorithms, including Iterative Single Data Algorithm (ISDA), L1-regularized Quadratic Programming (L1QP), and Sequential Minimal Optimization (SMO), were tested. Cross-validation on the training dataset identified the combination of kernel function and solver that achieved the highest mean accuracy and specificity.

**2.5.3. *Multilayer perceptron classifier*.** The MLP classifier, a type of feedforward neural network, was used to classify intact and defective PM stators. The network comprised an input layer, a single hidden layer, and an output layer. The hidden layer employed a hyperbolic tangent sigmoid activation function to introduce nonlinearity and improve pattern recognition, while the output layer used a linear activation function to support regression-based classification.

The training process aimed to minimize the mean squared error between predicted and actual class labels. Thirteen training algorithms were evaluated, including Levenberg-Marquardt (LM), Bayesian Regularization (BR), Scaled Conjugate Gradient (SCG), Broyden–Fletcher–Goldfarb–Shanno (BFGS), One-Step Secant (OSS), and several gradient descent methods. For each algorithm, the number of hidden neurons, learning rate, momentum, and weight update strategies were systematically optimized through iterative testing. Cross-validation was applied to ensure robust performance evaluation and to reduce the influence of variations in data partitions.

                                                                

Hyperparameter tuning identified an optimal network configuration. The number of hidden neurons was set to 15 based on cross-validation results. Learning rates and weight initialization schemes were adjusted for each algorithm to achieve stable convergence. The final training algorithm was selected by comparing mean accuracy, specificity, and overall performance across both training and test datasets. Mean squared error served as the primary criterion for performance during training.

**2.5.4. Radial basis function classifier.** The RBF neural network was applied to classify intact and defective PM stators. The network architecture included an input layer, a single hidden layer with radial basis activation functions, and a linear output layer. The hidden layer employed normalized radial basis functions to introduce nonlinearity and improve pattern discrimination.

The network was trained to minimize the mean squared error between predicted and actual class labels. Thirteen training algorithms were evaluated, including LM, BR, BFGS, several conjugate gradient variants (CGF, CGP, CGB, SCG), OSS, RP, and gradient descent-based methods (GD, GDA, GDM, GDX). For each algorithm, the number of hidden neurons and the spread constant were iteratively optimized using cross-validation to ensure robust learning and generalizable performance.

Hyperparameter tuning established an optimal configuration for the RBF network. The number of hidden neurons was set to 20 based on cross-validation results, and the spread constant was adjusted to 0.6 to balance local generalization with global smoothness. The training algorithm was selected by comparing mean accuracy, specificity, and overall performance across both training and test datasets. Mean squared error served as the primary performance metric during training.

## 2.6. Evaluation of classifier performance

The performance of the classifiers was evaluated using several metrics to measure their ability to correctly classify intact and defective PM stators. Classification results were summarized in a confusion matrix, which includes True Positives (TP), False Positives (FP), True Negatives (TN), and False Negatives (FN). These values were then used to calculate the following metrics: accuracy, precision, recall (sensitivity), F1-score, specificity, AUC, and Youden's Index (YI).

- *Accuracy* indicates the overall correctness of the classifier.

- *Precision* measures the proportion of correctly identified positive instances.

- *Recall (sensitivity)* reflects the classifier's ability to detect positive instances.

- *F1-score* provides a balance between precision and recall.

- *Specificity* evaluates performance on negative instances.

- *AUC* measures the classifier's ability to distinguish between classes.

- *Youden's Index* balances sensitivity and specificity.

These metrics provide a comprehensive assessment of classifier performance and allow comparisons across different models. The metrics were calculated using the following equations [29, 30]:

$$Accuracy(\%) = \frac{TP + TN}{TP + TN + FP + FN} \times 100 \tag{6}$$

$$Precision(\%) = \frac{TP}{TP + FP} \times 100 \tag{7}$$

$$Sensitivity(Recall) = \frac{TP}{TP + FN} \times 100 \tag{8}$$

$$F-score(\%) = \frac{2 \times Sensitivity \times Precision}{Sensitivity + Precision} \times 100 \tag{9}$$

$$Specificity(\%) = \frac{TN}{TN + FP} \times 100 \qquad (10)$$

$$AUC = \frac{1}{2} \times \left( \frac{TP}{TP + FN} + \frac{TN}{TN + FP} \right) \qquad (11)$$

$$YI = Sensitivity - (1 - Specificity) \qquad (12)$$

These calculations were used to compare classifier performance and assess their ability to differentiate intact from defective PM stators.

To ensure a reliable and generalizable evaluation, a 5-fold cross-validation scheme was applied instead of using a single random train–test split. The dataset was randomly divided into five equal parts with similar class distributions. In each round, four parts (about 80% of the data) were used for training, and the remaining part (about 20%) was used for testing. This process was repeated five times so that each part served once as the test set. In this way, every sample was included in both training and testing, which reduced the bias that might occur from relying on only one data split. To further limit the effect of random initialization, the entire cross-validation procedure was repeated four times with different random seeds. This resulted in 20 independent runs for each classifier. The performance metrics from these runs were then averaged, and their standard deviations were reported to capture variability across folds and random states. This strategy offers a more consistent and reproducible assessment of classifier performance.

## 3. Results and discussion

This section presents the study results for two classification scenarios. The first scenario is a binary classification, distinguishing between intact and defective oil pump stators manufactured by powder metallurgy (PM stators). The second scenario involves a multi-class classification, where defective stators are further categorized into three types: (i) cracked components (CC), (ii) components with chipping or tooth breakage (CTB), and (iii) completely fractured components (CFC). This creates a four-class problem, including one intact class and three defective classes. To achieve the best classification performance, the design and operational parameters of each classifier were first optimized. Then, the most relevant features for each scenario were identified to systematically assess feature importance. Finally, the performance of the classifiers was compared to determine the most effective approach for defect diagnosis. This structured methodology provides clear insights into how feature selection and classifier design affect defect detection accuracy.

### 3.1. Classifier parameter optimization

In this study, four classifiers—SVM, KNN, MLP, and RBF network—were used to classify intact and defective PM stators based on acoustic signal features. Because classifier performance depends strongly on parameter settings, the optimal values were determined first. The results of this optimization are presented in the following sections. After optimization, the classifiers were applied to PM stator classification, and their performance was compared.

3.1.1. **SVM parameter optimization.** The performance of the SVM classifier was assessed through cross-validation using various kernel functions and solvers to determine the most suitable configuration for PM stator classification. As presented in Table 1, both kernel choice and solver exerted a significant impact on mean classification accuracy and specificity across training and testing phases. Among the evaluated configurations, the RBF kernel combined with the SMO solver yielded the highest performance, achieving 99% accuracy and specificity. The consistently low standard deviation values further highlighted the stability of this configuration across multiple datasets. By contrast, the linear and polynomial kernels (degrees 2 and 3) resulted in lower testing accuracy, accompanied by relatively higher standard deviation values. These findings suggest that their performance was more sensitive to dataset variability, thereby reducing

**Table 1. Mean and standard deviation of SVM classifier performance across different kernel functions and solvers with cross-validation.**

| Kernel Function | Solver | Train | | Test | | Overall | | Performance |
|---|---|---|---|---|---|---|---|---|
| | | Accuracy | Specificity | Accuracy | Specificity | Accuracy | Specificity | |
| linear | ISDA | 0.96±0.07 | 0.95±0.09 | 0.80±0.09 | 0.77±0.07 | 0.93±0.07 | 0.92±0.07 | 0.27±0.07 |
| | L1QP | 0.96±0.08 | 0.96±0.07 | 0.90±0.07 | 0.88±0.08 | 0.95±0.10 | 0.95±0.09 | 0.66±0.10 |
| | SMO | 0.98±0.08 | 0.97±0.06 | 0.90±0.08 | 0.88±0.07 | 0.96±0.08 | 0.95±0.07 | 0.70±0.09 |
| Polynomial (Degree 2) | ISDA | 0.99±0.01 | 0.99±0.01 | 0.85±0.08 | 0.85±0.06 | 0.97±0.09 | 0.97±0.07 | 0.63±0.08 |
| | ISDA | 0.99±0.01 | 0.99±0.01 | 0.85±0.05 | 0.88±0.07 | 0.97±0.10 | 0.98±0.08 | 0.69±0.06 |
| | L1QP | 0.99±0.01 | 0.99±0.01 | 0.85±0.08 | 0.85±0.07 | 0.97±0.08 | 0.97±0.07 | 0.63±0.06 |
| Polynomial (Degree 3) | L1QP | 0.99±0.01 | 0.99±0.01 | 0.90±0.07 | 0.92±0.09 | 0.98±0.09 | 0.98±0.08 | 0.84±0.05 |
| | SMO | 0.99±0.01 | 0.99±0.01 | 0.90±0.07 | 0.92±0.07 | 0.98±0.06 | 0.98±0.07 | 0.84±0.09 |
| | SMO | 0.99±0.01 | 0.99±0.01 | 0.80±0.10 | 0.79±0.08 | 0.96±0.07 | 0.96±0.09 | 0.43±0.08 |
| RBF | ISDA | 1.00±0.00 | 1.00±0.00 | 0.95±0.09 | 0.94±0.07 | 0.99±0.05 | 0.99±0.08 | 0.98±0.09 |
| | L1QP | 1.00±0.00 | 1.00±0.00 | 0.95±0.08 | 0.94±0.08 | 0.99±0.07 | 0.99±0.07 | 0.98±0.05 |
| | SMO | 1.00±0.00 | 1.00±0.00 | 0.95±0.07 | 0.94±0.08 | 0.99±0.07 | 0.99±0.08 | 0.98±0.10 |

**Notes:** ISDA (Iterative Single Data Algorithm); L1QP (L1-regularized Quadratic Programming); SMO (Sequential Minimal Optimization).

robustness. The superior performance of the RBF-SMO configuration is attributable to its capacity to effectively capture complex, nonlinear decision boundaries. For all SVM models, the box constraint was fixed at 1, the kernel offset at 0.1, and the kernel scale at 1. Overall, the results demonstrate that the RBF-SMO approach provides a reliable and robust method for discriminating between intact and defective PM stators.

**3.1.2. *KNN parameter optimization*.** The KNN classifier was evaluated using cross-validation and different distance metrics to identify the best configuration for PM stator classification. Table 2 reports the mean and standard deviation of accuracy and specificity for training, testing, and overall performance. The results show that the choice of distance metric strongly influenced classifier performance. Manhattan and Minkowski distances achieved perfect classification, with 100% accuracy and specificity in all phases. This demonstrates their robustness in distinguishing intact and defective PM

**Table 2. Mean and standard deviation of KNN classifier performance across different distance metrics with cross-validation.**

| Distance Metric | Train | | Test | | Overall | | Performance |
|---|---|---|---|---|---|---|---|
| | Accuracy | Specificity | Accuracy | Specificity | Accuracy | Specificity | |
| Manhattan | 1.00±0.00 | 1.00±0.00 | 1.00±0.00 | 1.00±0.00 | 1.00±0.00 | 1.00±0.00 | 1.00±0.00 |
| Chebyshev | 0.99±0.01 | 0.99±0.01 | 0.95±0.07 | 0.94±0.08 | 0.99±0.10 | 0.99±0.09 | 0.97±0.08 |
| Correlation | 0.99±0.01 | 0.99±0.01 | 0.70±0.18 | 0.69±0.07 | 0.94±0.08 | 0.94±0.07 | 0.85±0.05 |
| Cosine | 0.99±0.01 | 0.99±0.01 | 0.75±0.08 | 0.75±0.06 | 0.95±0.09 | 0.95±0.07 | 0.87±0.07 |
| Euclidean | 0.99±0.01 | 0.99±0.01 | 0.95±0.05 | 0.94±0.07 | 0.99±0.10 | 0.99±0.08 | 0.97±0.06 |
| Hamming | 0.99±0.01 | 0.99±0.01 | 0.40±0.28 | 0.44±0.17 | 0.88±0.18 | 0.89±0.17 | 0.71±0.19 |
| Jaccard | 0.99±0.01 | 0.99±0.01 | 0.40±0.27 | 0.44±0.29 | 0.88±0.19 | 0.89±0.08 | 0.71±0.18 |
| Mahalanobis | 0.99±0.01 | 0.99±0.01 | 0.90±0.07 | 0.88±0.07 | 0.98±0.06 | 0.98±0.07 | 0.95±0.07 |
| Minkowski | 0.99±0.01 | 0.99±0.01 | 0.99±0.01 | 0.99±0.01 | 0.99±0.01 | 0.99±0.01 | 0.99±0.01 |
| Standardized Euclidean | 0.99±0.01 | 0.99±0.01 | 0.95±0.09 | 0.94±0.07 | 0.99±0.05 | 0.99±0.08 | 0.97±0.07 |
| Spearman | 0.89±0.17 | 0.90±0.17 | 0.55±0.18 | 0.54±0.08 | 0.82±0.17 | 0.83±0.17 | 0.72±0.17 |

Note: The results are based on a KNN classifier configured with tie-breaking set to "smallest", a bucket size of 10, exhaustive search, equal weighting, one neighbor, standardization enabled, and an exponent of 1.

stators. Euclidean, Chebyshev, and Standardized Euclidean metrics also performed well, reaching 99% overall accuracy. In contrast, Hamming, Jaccard, and Spearman metrics produced lower test accuracy, indicating less reliable classification. Correlation and Cosine metrics gave intermediate results, showing moderate sensitivity to dataset variations. The superior performance of Manhattan and Minkowski metrics is due to their ability to capture the geometric relationships between feature vectors in the multi-dimensional input space. Low standard deviations for these metrics indicate stable performance across different cross-validation partitions, while higher standard deviations for other metrics suggest that their results are more dependent on the specific training and testing splits. All KNN classifiers were configured with tie-breaking set to "smallest," bucket size of 10, exhaustive search, equal weighting, one nearest neighbor, standardization enabled, and an exponent of 1. Using these settings with cross-validation provided a robust and reliable assessment of the classifier's generalizability for PM stator defect detection.

**3.1.3. *MLP parameter optimization*.** The MLP classifier was evaluated using thirteen different training algorithms to determine the best configuration for distinguishing intact and defective PM stators. The network had a single hidden layer with 15 neurons and a hyperbolic tangent sigmoid activation function. Mean squared error was used as the performance criterion. Cross-validation was applied to ensure reliable evaluation and account for variations in different data partitions. Table 3 shows the mean and standard deviation of accuracy and specificity during training, testing, and overall phases for each algorithm. The LM algorithm achieved the highest performance, with perfect training metrics (accuracy and specificity of 1.00) and strong testing results (both 0.95). This led to an overall accuracy of 0.99 and a performance score of 1.00. BR also performed well, with testing accuracy and specificity of 1.00, although its training metrics were slightly lower (0.94 and 0.93). Other algorithms, including SCG, BFGS, OSS, and various gradient descent variants, showed lower overall performance. Scores ranged from 0.80 down to 0.32 for the CGP algorithm. These results indicate that the choice of training algorithm strongly affects MLP performance. Advanced optimization algorithms such as LM and BR are more effective for accurately detecting PM stator conditions.

**3.1.4. *RBF parameter optimization*.** The RBF network was assessed using thirteen different training algorithms to optimize its capacity for distinguishing between intact and defective PM stators. The network architecture consisted

**Table 3. Mean and standard deviation of MLP classifier performance across different training algorithms with cross-validation.**

| Training Algorithm | Train | | Test | | Overall | | Performance |
|---|---|---|---|---|---|---|---|
| | Accuracy | Specificity | Accuracy | Specificity | Accuracy | Specificity | |
| LM | 0.99±0.01 | 0.99±0.01 | 0.95±0.08 | 0.95±0.06 | 0.99±0.10 | 0.99±0.07 | 1.00±0.01 |
| BR | 0.94±0.08 | 0.93±0.06 | 1.00±0.01 | 1.00±0.07 | 0.95±0.08 | 0.95±0.08 | 0.94±0.07 |
| SCG | 0.90±0.08 | 0.90±0.07 | 0.90±0.05 | 0.90±0.07 | 0.90±0.09 | 0.90±0.07 | 0.80±0.06 |
| BFGS | 0.89±0.08 | 0.88±0.08 | 0.85±0.08 | 0.85±0.09 | 0.88±0.06 | 0.88±0.09 | 0.74±0.07 |
| OSS | 0.88±0.07 | 0.86±0.07 | 0.80±0.07 | 0.79±0.07 | 0.86±0.07 | 0.84±0.08 | 0.66±0.07 |
| GDM | 0.87±0.07 | 0.85±0.07 | 0.80±0.07 | 0.79±0.08 | 0.85±0.05 | 0.84±0.07 | 0.65±0.08 |
| CGF | 0.85±0.06 | 0.84±0.08 | 0.80±0.10 | 0.79±0.07 | 0.84±0.07 | 0.83±0.08 | 0.64±0.08 |
| GDA | 0.89±0.08 | 0.89±0.07 | 0.75±0.09 | 0.73±0.08 | 0.86±0.07 | 0.86±0.08 | 0.63±0.09 |
| CGB | 0.82±0.08 | 0.79±0.06 | 0.75±0.08 | 0.73±0.08 | 0.80±0.07 | 0.78±0.08 | 0.54±0.07 |
| GD | 0.85±0.08 | 0.84±0.07 | 0.70±0.07 | 0.69±0.07 | 0.82±0.09 | 0.81±0.05 | 0.54±0.05 |
| RP | 0.77±0.17 | 0.77±0.17 | 0.75±0.27 | 0.77±0.10 | 0.76±0.17 | 0.77±0.27 | 0.53±0.28 |
| GDX | 0.82±0.09 | 0.80±0.09 | 0.70±0.08 | 0.69±0.18 | 0.79±0.27 | 0.78±0.06 | 0.50±0.19 |
| CGP | 0.67±0.19 | 0.65±0.07 | 0.65±0.17 | 0.67±0.09 | 0.67±0.08 | 0.65±0.09 | 0.32±0.18 |

Note: LM (Levenberg-Marquardt), BR (Bayesian Regularization), SCG (Scaled Conjugate Gradient), BFGS (Broyden-Fletcher-Goldfarb-Shanno), OSS (One-Step Secant), GDM (Gradient Descent with Momentum), CGF (Conjugate Gradient Fletcher-Reeves), GDA (Gradient Descent Adaptive learning rate), CGB (Conjugate Gradient Powell-Beale), GD (Gradient Descent), RP (Resilient Backpropagation), GDX (Gradient Descent with variable learning rate), CGP (Conjugate Gradient with Polak–Ribiére).

of a single hidden layer with 20 neurons, a normalized radial basis transfer function, and a spread constant of 0.6. To ensure robustness and generalizability, cross-validation was employed across both training and testing datasets. Table 4 summarizes the mean and standard deviation of classification performance for the evaluated algorithms. The LM algorithm delivered the best performance, achieving perfect accuracy and specificity (1.00) in all phases. BR and BFGS also yielded near-optimal results, with training accuracies of 0.99 and 0.98, respectively, and flawless outcomes during testing. Among the conjugate gradient methods, including CGF, GP, CGB, and SCG, testing results were consistently perfect, though training accuracies were slightly lower (approximately 0.93–0.94). Their overall performance remained strong, with average values ranging from 0.94 to 0.95. By contrast, OSS and RP achieved only moderate performance, while gradient descent variants—such as GDX, GD, GDA, and GDM—exhibited more pronounced performance degradation. GDM produced the weakest results, with the lowest overall accuracy. These outcomes highlight the decisive influence of the training algorithm on the effectiveness of RBF networks in PM stator fault diagnosis. Advanced algorithms such as LM, BR, and BFGS consistently outperformed simpler gradient-based methods. Moreover, the application of cross-validation confirmed the reliability and generalizability of the results across different data partitions

### 3.2. *Overall classifier performance evaluation*

The classification performance of SVM, KNN, MLP, and RBF classifiers was evaluated across training, testing, and overall phases, using 80% of the data for training. As shown in Table 5, KNN and RBF achieved perfect performance, with all metrics—accuracy, specificity, precision, recall, F1-score, and AUC—equal to 1.00 in both training and testing phases. The SVM classifier showed slightly lower performance, with an overall accuracy of 0.99 and specificity of 0.98. Although SVM achieved near-perfect precision and recall, minor false positives occurred during training and testing. Similarly, the MLP classifier performed well, with an overall accuracy of 0.99, but recall and specificity slightly decreased in the test phase due to some false positives. The Youden Index highlights the robustness of the classifiers: KNN and RBF scored 1.00, while SVM and MLP scored slightly lower at 0.98. These results indicate that, although all classifiers performed well, KNN and RBF consistently provided higher reliability and overall robustness compared to SVM and MLP.

**Table 4. Mean and standard deviation of RBF classifier performance across different training algorithms with cross-validation.**

| Training Algorithm | Train | | Test | | Overall | | Performance |
|---|---|---|---|---|---|---|---|
| | Accuracy | Specificity | Accuracy | Specificity | Accuracy | Specificity | |
| LM | 1.00±0.00 | 1.00±0.00 | 1.00±0.00 | 1.00±0.00 | 1.00±0.00 | 1.00±0.00 | 1.00±0.00 |
| BR | 0.99±0.08 | 0.99±0.06 | 1.00±0.00 | 1.00±0.00 | 0.99±0.08 | 0.99±0.08 | 0.99±0.07 |
| BFGS | 0.98±0.08 | 0.97±0.07 | 1.00±0.00 | 1.00±0.00 | 0.98±0.09 | 0.98±0.07 | 0.98±0.06 |
| CGF | 0.94±0.08 | 0.93±0.08 | 1.00±0.00 | 1.00±0.00 | 0.95±0.06 | 0.94±0.09 | 0.94±0.07 |
| CGP | 0.94±0.07 | 0.93±0.07 | 1.00±0.00 | 1.00±0.00 | 0.95±0.07 | 0.94±0.08 | 0.94±0.07 |
| CGB | 0.94±0.07 | 0.92±0.07 | 1.00±0.00 | 1.00±0.00 | 0.95±0.05 | 0.94±0.07 | 0.94±0.08 |
| SCG | 0.93±0.06 | 0.92±0.08 | 1.00±0.00 | 1.00±0.00 | 0.94±0.07 | 0.94±0.08 | 0.93±0.08 |
| OSS | 0.91±0.08 | 0.90±0.07 | 0.95±0.09 | 0.94±0.08 | 0.92±0.07 | 0.91±0.08 | 0.91±0.09 |
| RP | 0.91±0.08 | 0.91±0.06 | 0.90±0.08 | 0.90±0.08 | 0.91±0.07 | 0.91±0.08 | 0.91±0.07 |
| GDX | 0.90±0.18 | 0.91±0.17 | 0.80±0.17 | 0.81±0.17 | 0.88±0.19 | 0.89±0.15 | 0.90±0.15 |
| GD | 0.88±0.27 | 0.85±0.17 | 0.85±0.17 | 0.81±0.10 | 0.87±0.27 | 0.84±0.27 | 0.88±0.28 |
| GDA | 0.84±0.29 | 0.80±0.19 | 0.80±0.18 | 0.75±0.18 | 0.83±0.27 | 0.79±0.26 | 0.84±0.29 |
| GDM | 0.82±0.19 | 0.80±0.17 | 0.70±0.17 | 0.69±0.29 | 0.79±0.28 | 0.78±0.29 | 0.82±0.28 |

Note: LM (Levenberg-Marquardt), BR (Bayesian Regularization), SCG (Scaled Conjugate Gradient), BFGS (Broyden-Fletcher-Goldfarb-Shanno),CGP (Conjugate Gradient Polak-Ribiere), OSS (One-Step Secant), GDM (Gradient Descent with Momentum), CGF (Conjugate Gradient Fletcher-Reeves), GDA (Gradient Descent Adaptive learning rate), CGB (Conjugate Gradient Powell-Beale), GD (Gradient Descent), RP (Resilient Backpropagation), GDX (Gradient Descent with variable learning rate).

**Table 5. Performance metrics of the SVM classifier across training, testing, and overall phases.**

| | Phase | Class | TP | FP | Acc | Prec | Rec | F1 | Spec | AUC | YI |
|---|---|---|---|---|---|---|---|---|---|---|---|
| SVM | Train | 1 | 31 | 0 | 0.99 | 1.00 | 0.97 | 0.98 | 1.00 | 0.98 | 0.97 |
| | | 2 | 50 | 1 | 0.99 | 0.98 | 1.00 | 0.99 | 0.97 | 0.98 | 0.97 |
| | Test | 1 | 8 | 0 | 1.00 | 1.00 | 1.00 | 1.00 | 1.00 | 1.00 | 1.00 |
| | | 2 | 12 | 0 | 1.00 | 1.00 | 1.00 | 1.00 | 1.00 | 1.00 | 1.00 |
| | Overall | 1 | 39 | 0 | 0.99 | 1.00 | 0.98 | 0.99 | 1.00 | 0.99 | 0.98 |
| | | 2 | 62 | 1 | 0.99 | 0.98 | 1.00 | 0.99 | 0.98 | 0.99 | 0.98 |
| | Total | | | | 0.99 | 0.99 | 0.99 | 0.99 | 0.99 | 0.99 | 0.99 |
| KNN | Train | 1 | 32 | 0 | 1.00 | 1.00 | 1.00 | 1.00 | 1.00 | 1.00 | 1.00 |
| | | 2 | 50 | 0 | 1.00 | 1.00 | 1.00 | 1.00 | 1.00 | 1.00 | 1.00 |
| | Test | 1 | 8 | 0 | 1.00 | 1.00 | 1.00 | 1.00 | 1.00 | 1.00 | 1.00 |
| | | 2 | 12 | 0 | 1.00 | 1.00 | 1.00 | 1.00 | 1.00 | 1.00 | 1.00 |
| | Overall | 1 | 40 | 0 | 1.00 | 1.00 | 1.00 | 1.00 | 1.00 | 1.00 | 1.00 |
| | | 2 | 62 | 0 | 1.00 | 1.00 | 1.00 | 1.00 | 1.00 | 1.00 | 1.00 |
| | Total | | | | 1.00 | 1.00 | 1.00 | 1.00 | 1.00 | 1.00 | 1.00 |
| MLP | Train | 1 | 32 | 0 | 1.00 | 1.00 | 1.00 | 1.00 | 1.00 | 1.00 | 1.00 |
| | | 2 | 50 | 0 | 1.00 | 1.00 | 1.00 | 1.00 | 1.00 | 1.00 | 1.00 |
| | Test | 1 | 8 | 1 | 0.95 | 0.89 | 1.00 | 0.94 | 0.92 | 0.96 | 0.92 |
| | | 2 | 11 | 0 | 0.95 | 1.00 | 0.92 | 0.96 | 1.00 | 0.96 | 0.92 |
| | Overall | 1 | 40 | 1 | 0.99 | 0.98 | 1.00 | 0.99 | 0.98 | 0.99 | 0.98 |
| | | 2 | 61 | 0 | 0.99 | 1.00 | 0.98 | 0.99 | 1.00 | 0.99 | 0.98 |
| | Total | | | | 0.99 | 0.99 | 0.99 | 0.99 | 0.99 | 0.99 | 0.98 |
| RBF | Train | 1 | 32 | 0 | 1.00 | 1.00 | 1.00 | 1.00 | 1.00 | 1.00 | 1.00 |
| | | 2 | 50 | 0 | 1.00 | 1.00 | 1.00 | 1.00 | 1.00 | 1.00 | 1.00 |
| | Test | 1 | 8 | 0 | 1.00 | 1.00 | 1.00 | 1.00 | 1.00 | 1.00 | 1.00 |
| | | 2 | 12 | 0 | 1.00 | 1.00 | 1.00 | 1.00 | 1.00 | 1.00 | 1.00 |
| | Overall | 1 | 40 | 0 | 1.00 | 1.00 | 1.00 | 1.00 | 1.00 | 1.00 | 1.00 |
| | | 2 | 62 | 0 | 1.00 | 1.00 | 1.00 | 1.00 | 1.00 | 1.00 | 1.00 |
| | Total | | | | 1.00 | 1.00 | 1.00 | 1.00 | 1.00 | 1.00 | 1.00 |

Abbreviations: TP: True Positives, FP: False Positives, Acc: Accuracy, Prec: Precision, Rec: Recall, F1: F1-Score, Spec: Specificity, AUC: Area Under the Curve, YI: Youden Index

Although all classifiers achieved high overall accuracy, the analysis by defect type revealed differences in detection performance. Defects with strong acoustic signatures, such as severe cracks, were consistently identified by all models without error. More subtle defects, including micro-porosity and shallow surface flaws, proved slightly more challenging and resulted in occasional false positives for the SVM and MLP classifiers. These findings indicate that models with stronger nonlinear mapping capabilities, such as KNN and RBF, are more effective in capturing the subtle spectral variations associated with less distinguishable defects. The performance gap can be explained by the greater sensitivity of SVM and MLP to small overlaps in feature space, whereas KNN and RBF preserve local neighborhood information and maintain nonlinear separability more effectively.

### 3.3. Impact of training set size on classifier generalization

The influence of training set size on the generalization capability of four classifiers—SVM, KNN, MLP, and RBF—was assessed using cross-validation (Fig 6). Classification accuracy was adopted as the primary performance metric to ensure a robust and reliable evaluation. The SVM classifier exhibited a pronounced dependency on training set size.

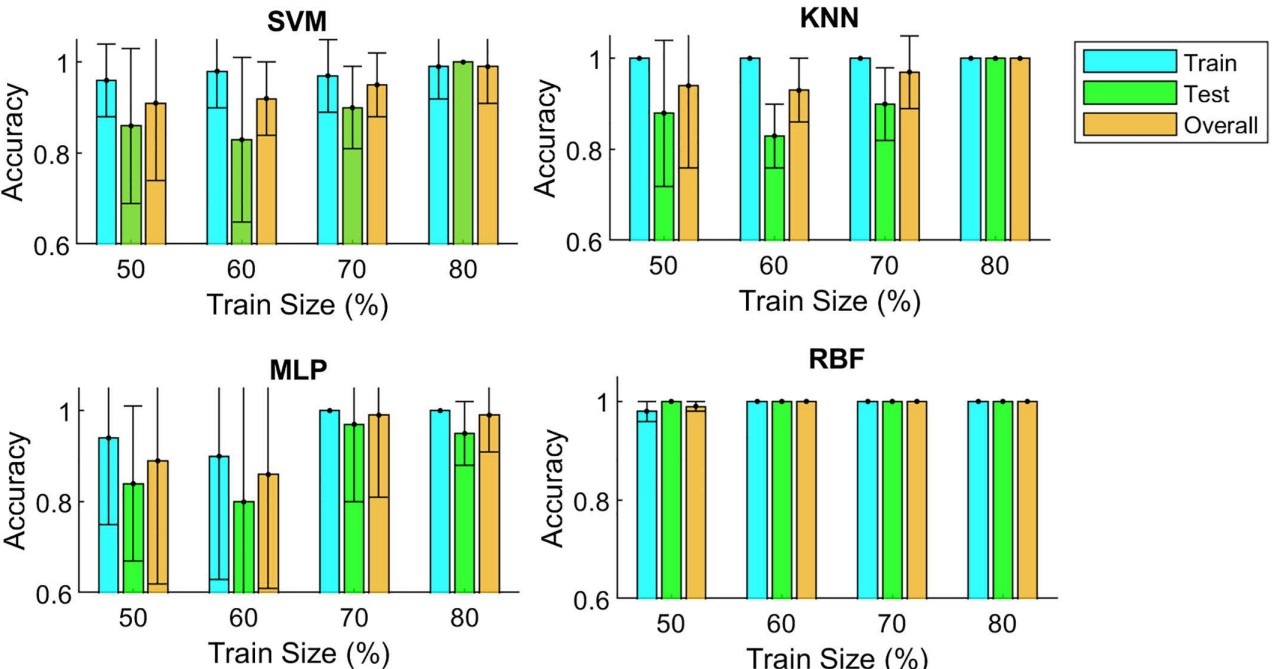

**Fig 6. Impact of training set size on the mean and standard deviation of SVM, KNN, MLP, and RBF classification accuracy under cross-validation.**

When 80% of the data was allocated for training, it achieved near-optimal accuracy (0.99 for training and 1.00 for testing). However, performance declined as the proportion of training data decreased, confirming the sensitivity of SVM to limited sample sizes. KNN consistently reached perfect training accuracy (1.00) across all training sizes. Its testing accuracy, although generally high, showed a gradual decline with reduced training data, reflecting a tendency toward overfitting under constrained datasets. The MLP classifier demonstrated strong performance with larger training sets (80% and 70%), but its accuracy deteriorated markedly when training data was reduced. Accuracy dropped to 0.86 at 60% and 0.89 at 50%, underscoring its limited robustness in scenarios with smaller datasets. In contrast, the RBF classifier achieved consistently superior outcomes. It maintained perfect or near-perfect accuracy in both training and testing phases, even with training proportions as low as 50%. This highlights RBF as the most stable and reliable model in terms of generalization. In summary, the results demonstrate that high training accuracy alone does not guarantee effective generalization. Training set size plays a decisive role, and its impact is classifier-dependent. Among the evaluated models, RBF exhibited the greatest robustness under cross-validation, whereas SVM and MLP were more adversely affected by reductions in training data.

### 3.4. Feature selection for optimal classification

Building on the high performance of the RBF neural network, we conducted a feature selection process to identify the most informative acoustic signal features while reducing input dimensionality. Table 6 summarizes the impact of different feature subsets on RBF performance, including training and testing accuracy, specificity, and overall performance.

Using all 20 features produced perfect results (1.00 across all metrics). However, performance varied when feature groups were considered individually. For example, Peak Frequency features (P1–P4) achieved an overall performance of 0.90, whereas Peak Amplitude features (PA1–PA4) reached 0.80. Spectral Centroid features (C1–C4) showed stronger

**Table 6. Impact of feature selection on the mean and standard deviation of RBF classifier performance across different feature subsets using cross-validation.**

| Variable | Train | | Test | | Overall | | Performance |
|---|---|---|---|---|---|---|---|
| | Accuracy | Specificity | Accuracy | Specificity | Accuracy | Specificity | |
| All | 1.00±0.00 | 1.00±0.00 | 1.00±0.00 | 1.00±0.00 | 1.00±0.00 | 1.00±0.00 | 1.00±0.00 |
| P1–P4 | 0.94±0.08 | 0.92±0.07 | 0.95±0.09 | 0.94±0.10 | 0.95±0.07 | 0.93±0.07 | 0.90±0.05 |
| PA1–PA4 | 0.80±0.08 | 0.82±0.07 | 0.80±0.08 | 0.81±0.08 | 0.80±0.08 | 0.82±0.07 | 0.80±0.08 |
| C1–C4 | 0.95±0.08 | 0.94±0.18 | 0.95±0.17 | 0.94±0.09 | 0.95±0.07 | 0.94±0.08 | 0.93±0.07 |
| MA1–MA4 | 0.80±0.17 | 0.77±0.27 | 0.60±0.27 | 0.58±0.15 | 0.76±0.19 | 0.74±0.18 | 0.51±0.18 |
| S1–S4. | 0.93±0.17 | 0.92±0.16 | 0.80±0.18 | 0.79±0.18 | 0.90±0.08 | 0.90±0.19 | 0.85±0.18 |
| P1, PA1, C1, A1, S1 | 0.93±0.06 | 0.93±0.07 | 0.85±0.07 | 0.83±0.19 | 0.91±0.07 | 0.91±0.07 | 0.87±0.07 |
| P2, PA2, C2, MA2, S2 | 0.83±0.18 | 0.80±0.17 | 0.90±0.16 | 0.88±0.06 | 0.84±0.28 | 0.81±0.15 | 0.78±0.17 |
| P3, PA3, C3, MA3, S3 | 0.98±0.08 | 0.98±0.09 | 1.00±0.00 | 1.00±0.00 | 0.98±0.08 | 0.98±0.08 | 0.95±0.08 |
| P4, PA4, C4, MA4, S4 | 0.91±0.08 | 0.91±0.07 | 0.80±0.07 | 0.79±0.05 | 0.89±0.08 | 0.88±0.09 | 0.86±0.06 |
| P3, S3, P4, MA4 | 1.00±0.00 | 1.00±0.00 | 1.00±0.00 | 1.00±0.00 | 1.00±0.00 | 1.00±0.00 | 1.00±0.00 |
| P3, S3, PA4, C4 | 1.00±0.00 | 1.00±0.00 | 1.00±0.00 | 1.00±0.00 | 1.00±0.00 | 1.00±0.00 | 1.00±0.00 |
| PA3, S3, P4, C4 | 1.00±0.00 | 1.00±0.00 | 1.00±0.00 | 1.00±0.00 | 1.00±0.00 | 1.00±0.00 | 1.00±0.00 |

**Note:** The feature set comprises 20 spectral features extracted from four frequency ranges (1,000–5,000 Hz, 5,000–10,000 Hz, 10,000–15,000 Hz, and 15,000–20,000 Hz). These include Peak Frequency (P1–P4), Peak Amplitude (PA1–PA4), Spectral Centroid (C1–C4), Mean Amplitude (MA1–MA4), and Skewness (S1–S4), where the number indicates the corresponding frequency range.

discriminative power with an overall performance of 0.93, while Mean Amplitude (MA1–MA4) performed poorly (0.51), and Skewness features (S1–S4) achieved 0.85.

Analysis of the four frequency ranges (1,000–5,000 Hz, 5,000–10,000 Hz, 10,000–15,000 Hz, 15,000–20,000 Hz) revealed differing contributions to classification. Features from the third range (10,000–15,000 Hz) exhibited the highest discriminative power, consistently yielding superior accuracy and specificity. The fourth range (15,000–20,000 Hz) followed closely, particularly when combined with the third range, sometimes achieving perfect performance. Features from the first and second ranges contributed less to overall performance, indicating that the most informative spectral and statistical properties are primarily in the third and fourth frequency bands.

A feature set from the third range, including P3, PA3, C3, MA3, and S3, achieved a classification performance of 0.95. This guided the search for an optimal four-feature combination. An exhaustive search among all 4,845 possible four-feature sets identified three top-performing combinations: (i) P3, S3, P4, MA4; (ii) P3, S3, PA4, C4; and (iii) PA3, S3, P4, C4. These combinations, mainly from the third and fourth frequency ranges, achieved perfect accuracy, specificity, and overall performance.

The selected features are closely related to the defect mechanisms in PM components. PA3 indicates the intensity of vibrations caused by cracks or fractures, with cracked components producing higher peak amplitudes at specific frequencies due to sudden material discontinuities. S3 reflects asymmetry in the amplitude distribution and is sensitive to irregularities such as chipping or tooth breakage. P4 corresponds to shifts in the natural frequency resulting from material loss or fractures, with fully fractured components showing distinct frequency peaks compared to intact parts. C4 represents the center of mass of the frequency spectrum, which shifts in the presence of structural defects, enabling differentiation among intact, cracked, chipped, and fully fractured components. Together, these features capture both amplitude and frequency characteristics associated with mechanical changes from different defect types, supporting accurate and reliable

classification of intact and defective PM stators. These results highlight the importance of targeted feature selection. A minimal yet informative feature subset can reduce computational cost while maintaining robust classification, enabling precise fault detection in PM stators.

Fig 7 presents the classification boundaries between intact and defective PM stators based on the four selected features (P3, P4, S3, and MA4). As the feature space is four-dimensional, the figure provides a two-dimensional approximation by plotting pairs of features while keeping the other features fixed at their mean values. The colored regions represent the classifier's decision zones, with blue indicating defective regions and brown indicating intact regions. This visualization highlights how the classifier partitions the feature space, demonstrating that the selected features effectively capture the differences between intact and defective stators and reflect the underlying patterns associated with various defect types.

### 3.5. Four-class classification performance

Previous results showed that the optimally configured RBF neural network could perfectly classify PM stators into two classes (intact and defective) using a four-feature subset (Scenario 1). However, identifying the specific type of defect is also important, so a second scenario was defined. In this scenario, PM stators were classified into four classes: one intact class and three defective subgroups—cracked components (20 samples), components with chipping or tooth breakage

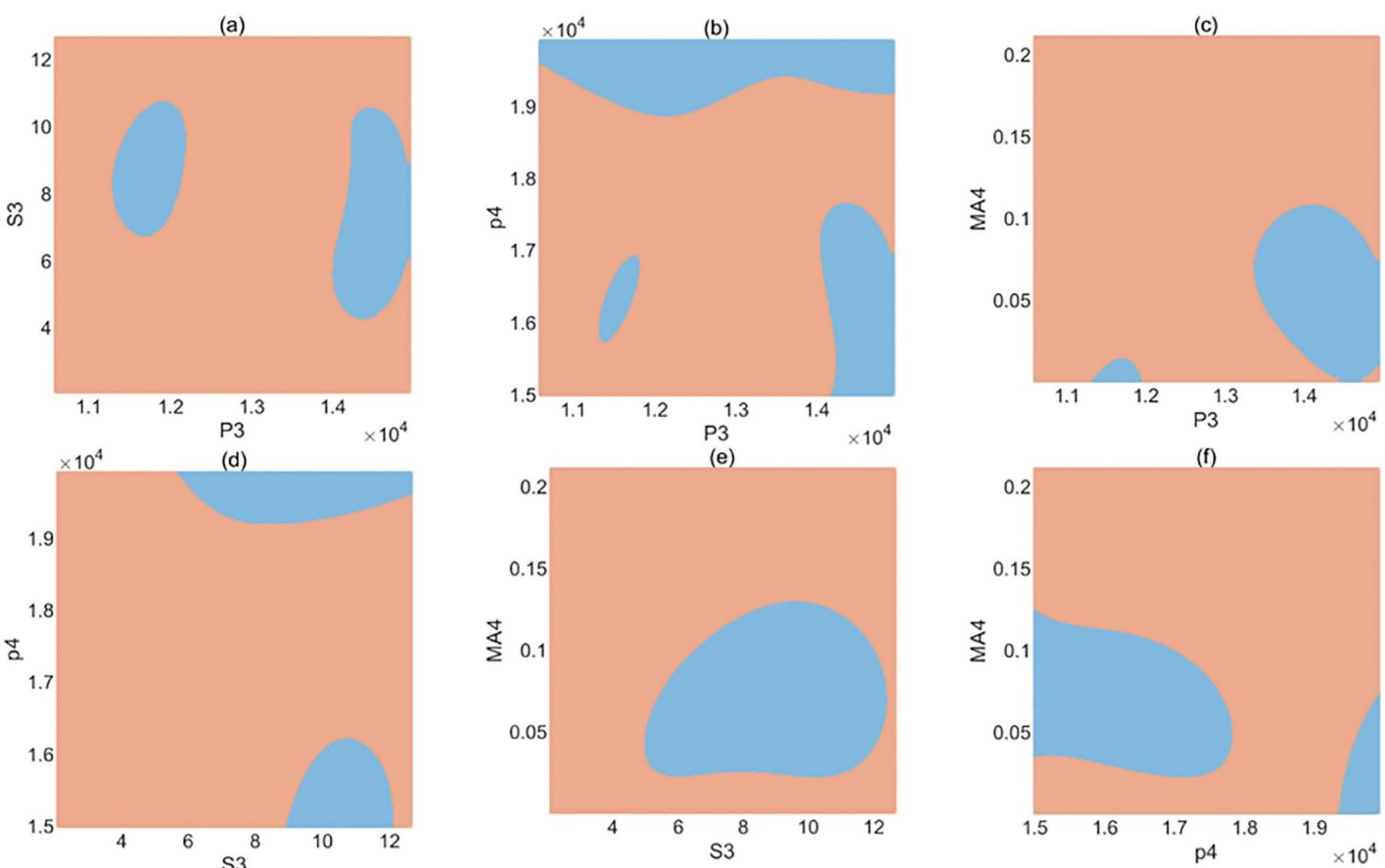

**Fig 7. Two-dimensional visualization of the classifier's decision boundaries for intact and defective PM stators using the selected features (P3, P4, S3, and MA4).**

(16 samples), and completely fractured components (26 samples). For this four-class problem, the same optimally configured RBF network was applied, and an exhaustive evaluation of all combinations of the 20 spectral features was conducted. The results are presented in Table 7.

Table 7 reports the classifiers' performance for the four-class problem, showing metrics for training, testing, and overall performance. The complete feature set (all 20 features) achieved perfect performance (1.00 across all metrics). However, evaluation of individual feature groups revealed notable differences. For example, the Peak Frequency features (P1–P4) achieved an overall performance of 0.89, while the Peak Amplitude subset (PA1–PA4) reached only 0.67. The Spectral Centroid features (C1–C4) and Skewness features (S1–S4) yielded overall performance values of 0.88 and 0.90, respectively. Moreover, combinations of features from specific frequency ranges—such as P3, PA3, C3, MA3, and S3—produced an overall performance of 0.94, whereas some combinations (e.g., PA3, S3, P4, C4) achieved near-perfect scores. Notably, the combination S2, PA3, C3, S3, P4, C4 attained perfect performance across all evaluation phases.

Interestingly, the subset previously used for perfect two-class classification—PA3, S3, P4, C4—showed a slightly lower overall performance (0.98) in the four-class scenario. By increasing the number of features to six, performance was fully restored; the combination S2, PA3, C3, S3, P4, C4 achieved flawless performance across training, testing, and overall evaluations. These features, mainly from the third (10,000–15,000 Hz) and fourth (15,000–20,000 Hz) frequency ranges, highlight the importance of high-frequency spectral and statistical properties in achieving robust fault classification of PM stators.

Table 8 provides a summary of recent advancements in integrating NDT methods with ML for defect detection across various industrial sectors. A notable trend is the widespread use of vision-based inspection (VBI) methods, applied in casting, welding, textile, and wood industries [31–41]. CNN-based architectures, including CAE, YOLOv5, and ensemble models, have achieved high detection accuracies, often above 95%. For instance, ACED reached 98.9% accuracy in metal casting [42], while YOLOv5 achieved 93.48% accuracy in hardwood defect detection [31]. Despite these successes,

**Table 7. Effect of feature selection on the mean and standard deviation of four-class classification performance across various feature subsets with cross-validation.**

| Variable | Train | | Test | | Overall | | Performance |
|---|---|---|---|---|---|---|---|
| | Accuracy | Specificity | Accuracy | Specificity | Accuracy | Specificity | |
| All | 1.00±0.00 | 1.00±0.00 | 1.00±0.00 | 1.00±0.00 | 1.00±0.00 | 1.00±0.00 | 1.00±0.00 |
| P1–P4 | 0.90±0.08 | 0.97±0.07 | 0.95±0.09 | 0.96±0.10 | 0.91±0.07 | 0.96±0.07 | 0.89±0.05 |
| PA1–PA4 | 0.73±0.08 | 0.90±0.07 | 0.59±0.08 | 0.86±0.08 | 0.69±0.08 | 0.89±0.07 | 0.67±0.08 |
| C1–C4 | 0.91±0.08 | 0.97±0.08 | 0.88±0.07 | 0.95±0.09 | 0.90±0.07 | 0.97±0.08 | 0.88±0.07 |
| MA1–MA4 | 0.56±0.07 | 0.89±0.07 | 0.60±0.07 | 0.81±0.10 | 0.67±0.09 | 0.87±0.08 | 0.65±0.08 |
| S1–S4. | 0.91±0.07 | 0.97±0.06 | 0.92±0.08 | 0.97±0.08 | 0.91±0.08 | 0.97±0.09 | 0.90±0.08 |
| P1, PA1, C1, A1, S1 | 0.90±0.06 | 0.96±0.07 | 0.83±0.07 | 0.93±0.09 | 0.88±0.07 | 0.95±0.07 | 0.84±0.07 |
| P2, PA2, C2, MA2, S2 | 0.83±0.08 | 0.95±0.07 | 0.83±0.06 | 0.94±0.06 | 0.84±0.08 | 0.94±0.05 | 0.79±0.07 |
| P3, PA3, C3, MA3, S3 | 0.95±0.08 | 0.98±0.09 | 1.00±0.00 | 1.00±0.00 | 0.96±0.08 | 0.97±0.08 | 0.94±0.08 |
| P4, PA4, C4, MA4, S4 | 0.89±0.08 | 0.94±0.07 | 0.89±0.07 | 0.94±0.05 | 0.88±0.08 | 0.94±0.09 | 0.83±0.06 |
| PA3, S3, P4, C4 | 0.97±0.07 | 0.99±0.08 | 1.00±0.00 | 1.00±0.00 | 0.98±0.05 | 0.99±0.08 | 0.98±0.06 |
| S2, PA3, C3, S3, P4, C4. | 1.00±0.00 | 1.00±0.00 | 1.00±0.00 | 1.00±0.00 | 1.00±0.00 | 1.00±0.00 | 1.00±0.00 |

**Note:** The feature set comprises 20 spectral features extracted from four frequency ranges (1,000–5,000 Hz, 5,000–10,000 Hz, 10,000–15,000 Hz, and 15,000–20,000 Hz). These include Peak Frequency (P1–P4), Peak Amplitude (PA1–PA4), Spectral Centroid (C1–C4), Mean Amplitude (MA1–MA4), and Skewness (S1–S4), where the number indicates the corresponding frequency range.

**Table 8. Overview of NDT methods and corresponding ML techniques for defect detection and classification in various industrial applications.**

| Application | NDT | ML Techniques | Objective | Best Accuracy | Selected Method | Ref |
|---|---|---|---|---|---|---|
| Microelectronics | AE | 1D/2D CNN, GoogleNet | Automated defect recognition; robustness to signal distortion | >99% | 2D CNN | [51] |
| Wire Arc Additive Manufacturing | AE | KNN, SVM, RF, CNN | Detection of defective bead segments | >87% | RF | [43] |
| Wall Paintings/ Bonded Structures | AE | CNN | Detection of debonding defects | 95% | CNN | [44] |
| Corrosion Monitoring | AE | Naïve Bayes, BP, RB | Detection and severity classification | 100% | RBF | [52] |
| Composite Tubes | AE | MLP, RF, SVM, NB, Clustering | Damage monitoring and classification | 99.24% | RF | [45] |
| Composite Materials | AE | SVM | Identification of damage mechanisms | 93.13% | SVM | [46] |
| Welding | IRT | ANN | Real-time identification of weld penetration states | >96% | ANN | [50] |
| Welding | IRT | ANN | Prediction of weld penetration depth | 88% | ANN | [49] |
| Oil & Gas Pipelines | MT | GA-KELM | Prediction of corrosion defect size | >98.8% | GA-KELM | [48] |
| Aeroengine Turbine Blades | RT | Unsupervised Adversarial Learning | Defect detection in X-ray images | 91.10% | Adversarial Learning | [47] |
| Welding | UT | CNN | Automated defect recognition (porosity, slag) | 85% | CNN | [42] |
| Oil & Gas | UT | EMD + WPD + RF | Defect identification and depth classification | 92.50% | RF | [53] |
| Metal Casting | VBI | CAE, CNN Ensemble | Automated detection of small surface defects; reduced false positives | 98.90% | ACED | [31] |
| Hardwood Flooring Manufacturing | VBI | YOLOv5 | Automated detection and classification of surface defects | 93.48% | YOLOv5 | [32] |
| Textile/ Packaging | VBI | CNN | Automated online defect detection in fabrics | 94% | CNN | [33] |
| Welding | VBI | CNN | Real-time weld penetration monitoring | 98.37% | CNN | [34] |
| Additive Manufacturing | VBI | CNN | Classification of powder bed defects | 95.80% | CNN | [35] |
| Permanent Magnet Motors | VBI | SVD | Crack defect detection | 94.29% | SVD | [36] |
| Furniture Manufacturing | VBI | CNN | Automated defect detection in edge-glued panels | 97% | CNN | [37] |
| Timber Construction | VBI | ICDW-YOLO | Crack detection in wooden materials | 79.02% | ICDW-YOLO | [38] |
| Casting Industry | VBI | CNN | Automated detection of casting defects | >98% | CNN | [39] |
| Underwater Inspection | VBI + UT | CNN, MLP, Rule-based Fusion | Detection of cracks, corrosion, and coating loss | 98.8–100% | Classifier Fusion | [40] |
| Bearing Fault Detection | Vibration & AE | SVM, LDA, MFE | Fault classification using vibration/acoustic signals | 98.28% | TSFDR-LDA | [41] |

*Abbreviations:* NDT methods include VBI – Vision-Based Inspection, IRT – Infrared Thermography, AE – Acoustic Emission Testing, VAE – Vibration and Acoustic Emission Testing, MT – Magnetic Testing, RT – Radiographic Testing, UT – Ultrasonic Testing, and VBI + UT – Hybrid Vision and Ultrasonic Testing. ML techniques include CNN – Convolutional Neural Network, 1D/2D CNN – One- or Two-Dimensional CNN, GoogleNet – GoogLeNet Architecture, KNN – K-Nearest Neighbors, SVM – Support Vector Machine, RF – Random Forest, MLP – Multi-Layer Perceptron, NB – Naïve Bayes, BP-NN – Backpropagation Neural Network, RBF – Radial Basis Function Neural Network, Clustering – Clustering Algorithms (Genetic K-means, Hierarchical), ANN – Artificial Neural Network, GA-KELM – Genetic Algorithm–Improved Kernel Extreme Learning Machine, EMD – Empirical Mode Decomposition, WPD – Wavelet Packet Decomposition, CAE – Convolutional Autoencoder, YOLOv5 – You Only Look Once version 5, SVD – Singular Value Decomposition, ICDW-YOLO – Improved Crack Detection in Wooden Materials using YOLO, LDA – Linear Discriminant Analysis, MFE – Multidomain Feature Extraction, and Rule-based Fusion – Decision-Level Fusion using a Rule-Based Classifier

vision-based techniques often require complex imaging setups, high-resolution acquisition systems, and significant computational resources, limiting their scalability for real-time, high-throughput manufacturing. Environmental factors such as lighting, surface contamination, and the accessibility of hidden defects may also reduce reliability.

In contrast, acoustic emission (AE) methods offer similar accuracy while providing operational advantages. Studies in corrosion monitoring and composite tubes achieved nearly perfect classification using RBF networks and Random Forests [43,44]. Likewise, 2D CNNs for microelectronics inspection exceeded 99% accuracy [29]. These results indicate that acoustic signal analysis can provide robust diagnostic information under varying noise conditions while requiring simpler and more portable instrumentation than imaging approaches. Moreover, AE inherently captures subsurface and internal defects, often inaccessible to optical systems.

Other NDT modalities—infrared thermography (IRT), ultrasonic testing (UT), radiography (RT), and magnetic testing (MT)—are also widely applied. ANN-based IRT in welding achieved >96% accuracy for penetration state classification [45], though penetration depth prediction remained lower (88%) [46], showing the difficulty of capturing continuous defect parameters. UT methods that combine signal decomposition (EMD, WPD) with RF classifiers reached up to 92.5% accuracy [47], whereas CNN-based ultrasonic recognition was lower at 85% [48]. These results suggest that hybrid signal processing and feature-engineered methods often outperform deep learning when datasets are small or defects are subtle. RT applied to aeroengine blade inspection reached 91.1% accuracy with adversarial learning [49], highlighting potential but also the computational demands of imaging-based methods. MT techniques, such as GA-KELM, achieved 98.8% accuracy in corrosion defect sizing [50], showing the promise of optimization-driven approaches in traditional NDT.

Multimodal and hybrid approaches further demonstrate the benefits of combining complementary data sources. For example, underwater inspections using VBI and UT achieved 98.8–100% accuracy through classifier fusion [38], showing that integration can overcome limitations of single modalities. Similarly, combining vibration and AE signals for bearing fault detection reached 98.28% accuracy with TSFDR-LDA [39], confirming the diagnostic value of acoustic and vibrational data for rotating machinery.

Compared to these studies, this work demonstrates the effectiveness of acoustic resonance testing with ML for PM components. Unlike vision-based or radiographic methods, acoustic resonance testing does not require complex imaging systems or controlled environments. Instead, it uses material vibration characteristics, allowing rapid, low-cost, and non-invasive inspection suitable for industrial production. While previous AE studies reported high accuracies in specific fields [29,30,43,44,51,52], this work extends acoustic signal analysis to PM components, where defect detection has mostly relied on visual inspection. Importantly, the proposed method achieved 100% accuracy in both binary and four-class classification, surpassing most imaging- and UT-based approaches listed in Table 8. Additionally, the robustness of the RBF model with limited training data demonstrates its potential for industrial scalability, where labeled datasets are often small.

Overall, this comparison shows that although imaging-based NDT dominates the literature and performs well in controlled settings, its operational complexity limits practical deployment. Acoustic methods, especially those using resonance signals as in this study, combine simplicity, portability, and high diagnostic accuracy, making them strong candidates for real-time defect detection in PM and other materials. This contribution not only validates the industrial relevance of acoustic resonance testing but also expands the use of AE-based NDT to a new class of materials and manufacturing challenges.

## Conclusions

This study demonstrated that acoustic resonance testing combined with ML classifiers can accurately detect defects in powder metallurgy components. The main findings are summarized as follows:

- SVM achieved 99% accuracy, KNN reached 100% accuracy, and both MLP and RBF classifiers showed near-perfect training and testing performance. These results demonstrate the robustness and reliability of the models.

- KNN and RBF achieved perfect overall performance, while SVM and MLP reached approximately 0.99 accuracy with minor false positives. RBF maintained near-perfect generalization using only 50% of the training data. In contrast, SVM and MLP were more sensitive to smaller datasets, and KNN showed slight decreases in testing accuracy.

- Carefully chosen RBF features from the third and fourth frequency ranges enabled perfect classification, including four-class distinction. This shows that dimensionality can be reduced without losing robustness and highlights the importance of mid-to-high frequency features for distinguishing defect types.

- Acoustic resonance testing combined with ML achieved 100% accuracy, outperforming other non-destructive methods. This approach enables reliable quality control and fault diagnosis of PM components using 10–20 kHz frequency bands.

In conclusion, combining acoustic resonance testing with ML provides a robust, efficient, and scalable approach for inspecting PM components. Although the method is highly accurate, it does not provide information on defect location or severity, which limits its applicability in industrial decision-making. Future work should focus on developing methods for defect localization and severity estimation, implementing automated feature selection, enabling real-time analysis, and extending the approach to other industrial components.

## Author contributions

**Conceptualization:** Hamid Moeenfard.

**Data curation:** Mohammad Hossein Hadizadeh Isfahani.

**Formal analysis:** Mohammad Hossein Hadizadeh Isfahani, Abbas Rohani.

**Software:** Abbas Rohani.

**Supervision:** Hamid Moeenfard.

**Writing – original draft:** Mohammad Hossein Hadizadeh Isfahani, Abbas Rohani.

**Writing – review & editing:** Abbas Rohani.

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
