## [Decision Letter · Decision Letter 0]

21 Jul 2025

Dear Dr. Rohani,

Thank you for submitting your manuscript to PLOS ONE. After careful consideration, we feel that it has merit but does not fully meet PLOS ONE’s publication criteria as it currently stands. Therefore, we invite you to submit a revised version of the manuscript that addresses the points raised during the review process.

We look forward to receiving your revised manuscript.

Kind regards,

Mohammad Azadi

Academic Editor

PLOS ONE

Journal Requirements: 

3. Thank you for uploading your study's underlying data set. Unfortunately, the repository you have noted in your Data Availability statement does not qualify as an acceptable data repository according to PLOS's standards.

Additional Editor Comments:

The manuscript must be revised based on the reviewers’ comments plus the following issues,

1) A separated file must be provided for the authors’ answers to the comments, one by one. Moreover, all changes must be yellow-colored highlighted sentences in the revised article. The track changes condition is not suggested.

2) No abbreviations should be used in the keywords. Moreover, they must be also found in the abstract or the title.

3) The introduction is lengthy. Only 3 pages are enough. Moreover, the novelty of the manuscript must be highlighted in the introduction, compared to the literature review.

4) All formulations need references, unless they were extracted or introduced by the authors.

5) The scale bar must be provided for macroscopic and microscopic images, such as Fig. 1.

6) Generally, the discussion is poor. The obtained results must be described firstly and then; they must be compared to the other results of other articles. This technical issue could be also found from the low number of references.

7) The number of references must be extended to 35-40 references, at least, based on recent published articles in 2020-2025.

8) “Conclusion” must be changed to “Conclusions”.

Reviewers' comments:

Reviewer's Responses to Questions

**Comments to the Author**

1. Is the manuscript technically sound, and do the data support the conclusions?

Reviewer #1: Yes

Reviewer #2: Yes

2. Has the statistical analysis been performed appropriately and rigorously?

Reviewer #1: Yes

Reviewer #2: Yes

3. Have the authors made all data underlying the findings in their manuscript fully available?

Reviewer #1: Yes

Reviewer #2: Yes

4. Is the manuscript presented in an intelligible fashion and written in standard English?

Reviewer #1: No

Reviewer #2: Yes

Reviewer #1: Dear Editor,

I recommend the paper for major revision with the following comments:

1: The title of the paper is too general, especially since the authors mention “powder metallurgy components” without specifying the material. This approach overgeneralizes the results obtained from a limited number of samples to the entire manufacturing process.

2: You should avoid using overly polished or complicated text in a research paper. For example, the sentence “Furthermore, PM manufacturing typically involves high production volumes, necessitating not only accurate defect detection but also rapid inspection methods” is unnecessarily complex. It uses correlative conjunctions within a present participle clause, which makes it difficult to follow. Also, avoid combining phrases like “due to” and “however” in one sentence. It would be clearer to restructure the sentence using a single transition word such as “although.” Initially, I mentioned this issue in the abstract, but after reviewing the rest of the paper, I recommend checking the entire text. Although the language is grammatically correct, it is sometimes hard to read. A research paper should use clear, direct, and consistent structures — it should not include a variety of sentence forms like TOEFL or IELTS essays.

3: The advantage of using machine learning (ML) in this research is not mentioned in the abstract. It should be explained how this method can be effective and how it compares with other conventional methods. Currently, the abstract only highlights the necessity of using acoustic signal testing, without emphasizing the role or benefits of ML. Additionally, instead of providing a long general introduction in the abstract, it would be more useful to include details about the algorithm training process. Overall, the abstract needs to be completely restructured.

4: How is it possible that SVM and MLP neural network are listed as keywords in your paper but are not mentioned in the title or abstract? These important terms should at least appear in the abstract to reflect the main methods used in the study.

5: Please avoid introducing abbreviations for the first time without proper definition. When using an abbreviation, it should be introduced once with its full form, and then only the abbreviation should be used throughout the paper. For example, you have introduced “PM” and “ML” several times in different sections. Please check and correct this issue for all abbreviations in the text.

6: In the introduction, several long sentences make it difficult to follow the ideas clearly. The sentence structures are similar to those typically found in TOEFL reading passages, which are unnecessarily complex for a research paper. I recommend simplifying and restructuring these sentences to improve clarity and readability.

7: The introduction is too long, and the ideas in the paragraphs are not well connected. As a result, it is difficult for the reader to follow the overall direction of the paper. Moreover, the extended explanation about different types of non-destructive inspection methods makes the introduction unnecessarily lengthy and somewhat boring. In particular, the second paragraph does not contribute to the main objective of the paper. Instead, it seems to list information without offering a clear link to the research focus. Therefore, I suggest reorganizing and shortening the introduction to improve clarity, relevance, and reader engagement.

8: The long explanation of vibration non-destructive testing in paragraph 3 of the introduction may be boring for the reader. It would be better to shorten this section and focus more on information directly relevant to the study.

9: The literature review in the introduction related to acoustic testing is insufficient. Instead of listing different types of quality control methods and inspections in paragraph 2 and providing a long introduction to vibration testing at the beginning of paragraph 3, please include more detailed information about how acoustic testing differs from other methods. Also, explain the necessity of using this method compared to others, recent developments in acoustic testing, and its advantages.

10: Before the last paragraph of the introduction, which presents the novelty of the paper, the authors should clearly elaborate on the research gap and the necessity of conducting this study. Specifically, the following questions need to be addressed in detail:

- What are the necessities of using machine learning classification in this context?

- What advantages does the proposed method offer over existing approaches?

I recently reviewed a paper for PLOS One that used classification for defect detection in aluminum samples. I am interested to know whether changing the materials, manufacturing methods, or machine learning algorithms can contribute to solving real-world problems. If this method provides new insights or tools for researchers, the authors should explain this clearly in the research gap section of their paper.

11: At the end of the introduction, the authors should provide the reader with a clear summary of the novelty and methodology. This should include how the samples were prepared, how the algorithms were trained, and whether different algorithms were compared to find the best one. It should also explain how the accuracy of the models was evaluated, if train-test splitting was used to assess the ability to predict new samples, and what materials were tested. All these details should be clearly defined in the last paragraph of the introduction.

12: The methodology section of the paper is confusing and lacks a clear summary. I suggest the authors include a detailed flowchart that outlines the sequence of steps taken in the study. This would greatly improve the readability and help readers understand the research process more quickly.

13: When reporting a case study in a scientific paper, simply naming the very reliable vehicle (Tiba) equipment is not sufficient. You should provide detailed information such as the manufacturer, the factory, and the years of production. For example, in Iran, several companies manufacture similar equipment, so if it is not from Saipa, that should be clearly stated. All relevant details must be included to give full context. Moreover, the authors should explain why defect detection is particularly necessary for this type of equipment.

14: How did the defects occur in the case studies? Were similar types of defects observed among the samples?

15: The train-test splitting should not depend on only one specific random state. How can you generalize your model’s accuracy to other random states? I suggest using a “for” loop to change the random state multiple times and then report the mean values of all model metrics for both training and testing sets. Similar approaches were used in works such as “Explainable Artificial Intelligence Modeling of Internal Arc in a Medium Voltage Switchgear Based on Different CFD Simulations” and “Interpretable Machine Learning Modeling of Temperature Rise in a Medium Voltage Switchgear Using Multiphysics CFD Analysis.” In these studies, the mean metric values across different random states were reported, which improves the generalization of the models.

16: The formulation of the kernel function should be included in the paper. What about other kernel types? Also, did you use kernel functions with KNN? This method can achieve good results—for example, see “A Novel Machine Learning-Based Model for Predicting the Transition Fatigue Lifetime in Piston Aluminum Alloys.”

17: It is recommended to show the effect of training set size using figures and charts. For example, in “Modeling Nonlinear Deformation in Magnetic Polyelectrolyte Hydrogels: A Hybrid FEM-Machine Learning Framework,” there are figures titled “learning curves” that illustrate how the size of the dataset affects the model’s accuracy.

18: For writing the conclusion, please present the main findings of the paper using bullet points. The conclusions should include both qualitative and quantitative results to clearly summarize the key outputs.

Reviewer #2: In this study, a defect detection method based on acoustic signal and machine learning is proposed, which has important engineering application value, reasonable research method, rigorous experimental design, and the results are innovative and practical. However, there is still room for improvement in the depth of the literature review, the completeness of the experimental details, the theoretical support of the analysis of the results, and the discussion of the limitations of the research. It is recommended that the author revise and improve the following issues:

1.“Page 3”

The introduction provides a comprehensive introduction to the existing non-destructive testing techniques, but there is little description of acoustic resonance testing (ART).

For powder metallurgy materials, the non-destructive testing technology used in the existing research is not introduced, and it is suggested to supplement and enhance the necessity and innovation of the research.

2.“Page 10”

The training process of machine learning models is not fully described in the experimental part, please supplement the specific process and standards of hyperparameter optimization of each model.

3.“Page 13”

The excitation method is an electromagnetic exciter, but the magnitude of the excitation force, the position of the application point and the repeatability test results are not specified.

4.“Page 14”

Supplemental Fig. 3 for the longitudinal axis units.

5.“Page 19”

The comparison of model performance (such as SVM, KNN, MLP, RBF) is comprehensive, but the analysis of the difference in detection difficulty of different defect types is insufficient, and the differences in the performance of models on different defect types and the reasons are not deeply discussed.

6.“Page 26”

In the feature selection section, the optimal combination of features (e.g., PA3, S3, P4, C4, etc.) is determined, but the association of these features with the defect mechanism is not explained.

7.“Page 27”

Explain in detail how Fig. 5 distinguishes between intact and defective classification boundaries.

8.“Page 29”

The conclusions summarize the research results, but the limitations of the studies are not well discussed.

9.In the Results and Discussion section, it is recommended to cite the research results of others for discussion to support the research in this paper.

**Do you want your identity to be public for this peer review?** For information about this choice, including consent withdrawal, please see our Privacy Policy

Reviewer #1: No

Reviewer #2: No

---

## [Author Response · Author response to Decision Letter 1]

29 Aug 2025

Manuscript Title:

Non-destructive defect detection in powder metallurgy components using acoustic signals and machine learning classification

Manuscript PONE-D-25-28300R1

Dear Editor in Chief,

Thank you for taking the time to evaluate the suitability of our paper for your journal. The suggestions offered by the reviewers have been quite helpful. We have incorporated the suggested edits and have highlighted them in the main text. In addition, we also answered the questions posed by the reviewers point by point in the following sheet. We sincerely thank the reviewers for their efforts and appreciate their help in improving the overall quality of the work.

Kind regards,

Abbas Rohani

Additional Editor Comments:

The manuscript must be revised based on the reviewers’ comments plus the following issues, 1) A separated file must be provided for the authors’ answers to the comments, one by one. Moreover, all changes must be yellow-colored highlighted sentences in the revised article. The track changes condition is not suggested.

Answer: We sincerely appreciate your guidance. The manuscript has been carefully revised according to the reviewers’ comments. A separate file containing our point-by-point responses to all comments has been prepared. In addition, all modifications in the revised manuscript are highlighted in yellow, as requested.

2) No abbreviations should be used in the keywords. Moreover, they must be also found in the abstract or the title.

Answer: The keywords have been revised to remove abbreviations, and they are now consistent with the terms used in the title and the abstract

3) The introduction is lengthy. Only 3 pages are enough. Moreover, the novelty of the manuscript must be highlighted in the introduction, compared to the literature review.

Answer: The introduction has been shortened to within three pages, and the novelty of the manuscript has been clearly emphasized in comparison with the existing literature.

4) All formulations need references, unless they were extracted or introduced by the authors.

Answer: All formulations in the manuscript have been properly referenced.

5) The scale bar must be provided for macroscopic and microscopic images, such as Fig. 1.

Answer: The scale bar has now been added to all images, including Fig. 1. In addition, the outer diameter of the stators (5.00 ± 0.05 cm) has been specified in the figure caption to provide precise dimensional information

6) Generally, the discussion is poor. The obtained results must be described firstly and then; they must be compared to the other results of other articles. This technical issue could be also found from the low number of references.

Answer: We sincerely appreciate your valuable comment. Following your suggestion, Table 10 has been added at the end of the manuscript to summarize recent studies in the field of fault diagnosis using imaging, acoustic analysis, and related approaches. This addition not only highlights the position of the present study within the broader body of research but also provides a clear comparison with previous works, thereby strengthening the depth and comprehensiveness of the discussion section.

7) The number of references must be extended to 35-40 references, at least, based on recent published articles in 2020-2025.

Answer: Thank you for your recommendation. In line with your earlier suggestion, the number of up-to-date references relevant to the scope of this research has been increased, particularly by including studies published between 2020 and 2025.

8) “Conclusion” must be changed to “Conclusions”.

Answer: The section title has been revised from “Conclusion” to “Conclusions” accordingly.

Reviewer # 1

I recommend the paper for major revision with the following comments:

1: The title of the paper is too general, especially since the authors mention “powder metallurgy components” without specifying the material. This approach overgeneralizes the results obtained from a limited number of samples to the entire manufacturing process.

Answer: Thank you for your valuable comment. We agree with your observation that the previous title was too general. To address this concern and to more accurately reflect the specific scope of our study, we have revised the title to:

“Non-destructive defect detection in powder metallurgy automotive oil pump stators using acoustic signals and machine learning classification.”

This revision avoids overgeneralization and specifies the exact type of component investigated, ensuring that the results are clearly contextualized within automotive oil pump stators manufactured by powder metallurgy

2: You should avoid using overly polished or complicated text in a research paper. For example, the sentence “Furthermore, PM manufacturing typically involves high production volumes, necessitating not only accurate defect detection but also rapid inspection methods” is unnecessarily complex. It uses correlative conjunctions within a present participle clause, which makes it difficult to follow. Also, avoid combining phrases like “due to” and “however” in one sentence. It would be clearer to restructure the sentence using a single transition word such as “although.” Initially, I mentioned this issue in the abstract, but after reviewing the rest of the paper, I recommend checking the entire text. Although the language is grammatically correct, it is sometimes hard to read. A research paper should use clear, direct, and consistent structures — it should not include a variety of sentence forms like TOEFL or IELTS essays.

Answer: We have carefully reviewed the entire manuscript and revised the text wherever possible to simplify complex sentences, improve clarity, and ensure direct and consistent expression throughout the paper.

3: The advantage of using machine learning (ML) in this research is not mentioned in the abstract. It should be explained how this method can be effective and how it compares with other conventional methods. Currently, the abstract only highlights the necessity of using acoustic signal testing, without emphasizing the role or benefits of ML. Additionally, instead of providing a long general introduction in the abstract, it would be more useful to include details about the algorithm training process. Overall, the abstract needs to be completely restructured.

Answer: We sincerely thank the Reviewer for this valuable comment. In the revised version, the Abstract has been improved to explicitly highlight the advantages of machine learning over conventional threshold-based inspection techniques, emphasizing its ability to detect complex, non-linear acoustic signal patterns that are often missed by traditional approaches. In addition, the training process of the algorithms has been clarified by specifying that time- and frequency-domain features were extracted from the collected acoustic data and used to train and validate four supervised ML classifiers (SVM, k-NN, MLP, and RBF). Furthermore, the comparative results of these models have been briefly summarized to demonstrate their performance, with the RBF model achieving 100% accuracy.

4: How is it possible that SVM and MLP neural network are listed as keywords in your paper but are not mentioned in the title or abstract? These important terms should at least appear in the abstract to reflect the main methods used in the study.

Answer: We appreciate the Reviewer’s insightful observation. In accordance with your suggestion and the Editor’s guidance, the list of keywords has been carefully revised, and the Abstract has been rewritten to ensure that all major methods, including SVM and MLP, are explicitly mentioned. This issue has been fully addressed in the revised manuscript, and the updated list of keywords can be reviewed for confirmation

5: Please avoid introducing abbreviations for the first time without proper definition. When using an abbreviation, it should be introduced once with its full form, and then only the abbreviation should be used throughout the paper. For example, you have introduced “PM” and “ML” several times in different sections. Please check and correct this issue for all abbreviations in the text.

Answer: We sincerely thank the Reviewer for this valuable comment. Following your suggestion, we have carefully revised the manuscript to ensure that each abbreviation is introduced only once with its full form at the first appearance, and then consistently used throughout the text. Specifically, abbreviations such as “PM” and “ML” have been corrected accordingly. This revision has been applied uniformly across all sections of the manuscript.

6: In the introduction, several long sentences make it difficult to follow the ideas clearly. The sentence structures are similar to those typically found in TOEFL reading passages, which are unnecessarily complex for a research paper. I recommend simplifying and restructuring these sentences to improve clarity and readability.

Answer: We appreciate the reviewer’s valuable comment. Following the editor’s guidance to reduce the introduction to a maximum of three pages, this section has been revised accordingly. In addition, the issue rightly noted by the reviewer regarding long and complex sentence structures has been carefully addressed, not only in the introduction but also throughout other sections of the manuscript to enhance clarity and readability.

7: The introduction is too long, and the ideas in the paragraphs are not well connected. As a result, it is difficult for the reader to follow the overall direction of the paper. Moreover, the extended explanation about different types of non-destructive inspection methods makes the introduction unnecessarily lengthy and somewhat boring. In particular, the second paragraph does not contribute to the main objective of the paper. Instead, it seems to list information without offering a clear link to the research focus. Therefore, I suggest reorganizing and shortening the introduction to improve clarity, relevance, and reader engagement.

Answer: We sincerely appreciate your valuable feedback. In line with your recommendation and the editor’s suggestion, the introduction has been thoroughly revised and reorganized. The section was shortened to improve clarity, eliminate redundancies, and strengthen the logical flow of ideas. The part concerning non-destructive inspection methods was also streamlined to ensure relevance to the research objectives. We hope that these revisions adequately address your concern.

8: The long explanation of vibration non-destructive testing in paragraph 3 of the introduction may be boring for the reader. It would be better to shorten this section and focus more on information directly relevant to the study.

Answer: We appreciate your observation. This section has been shortened and simplified to avoid unnecessary detail. The primary aim of including it was to introduce the subject and highlight the importance of applying machine learning methods in this research area. In line with your valuable feedback, we have revised and restructured this part to improve clarity and relevance.

9: The literature review in the introduction related to acoustic testing is insufficient. Instead of listing different types of quality control methods and inspections in paragraph 2 and providing a long introduction to vibration testing at the beginning of paragraph 3, please include more detailed information about how acoustic testing differs from other methods. Also, explain the necessity of using this method compared to others, recent developments in acoustic testing, and its advantages.

Answer: We appreciate your comment regarding the coverage of acoustic testing in the introduction. The introduction has been thoroughly revised and streamlined to avoid unnecessary elaboration. However, based on our review and the available references, there is very limited research on defect detection in industrial components produced by powder metallurgy, particularly in the automotive sector. Therefore, to highlight the importance of acoustic testing, we referred to related studies where applicable, in order to clarify its relevance and advantages compared to other methods. Additionally, Table 10 has been added to the manuscript, providing a comprehensive overview of the literature in this area. As shown in the table, most existing studies focus on defect detection using image-based techniques, which further emphasizes the novelty and significance of our work in applying acoustic testing for industrial defect detection.

10: Before the last paragraph of the introduction, which presents the novelty of the paper, the authors should clearly elaborate on the research gap and the necessity of conducting this study. Specifically, the following questions need to be addressed in detail:

- What are the necessities of using machine learning classification in this context?

- What advantages does the proposed method offer over existing approaches?

I recently reviewed a paper for PLOS One that used classification for defect detection in aluminum samples. I am interested to know whether changing the materials, manufacturing methods, or machine learning algorithms can contribute to solving real-world problems. If this method provides new insights or tools for researchers, the authors should explain this clearly in the research gap section of their paper.

Answer: We appreciate the reviewer’s valuable comment. In accordance with your suggestion, the necessity of using machine learning classification in this context and the advantages of the proposed method compared to existing approaches have been clarified at the end of the introduction. In particular, the study by Han et al. [Han, Ying, Xingkun Li, Gongxiang Cui, Jie Song, Fengyu Zhou, and Yugang Wang. Multi-defect detection and classification for aluminum alloys with enhanced YOLOv8. PloS One, 20(3), e0316817, 2025] was reviewed. As also highlighted in our introduction, nearly all of these studies are image-based, which suffer from certain drawbacks such as being time-consuming, costly, and lacking sufficient speed for continuous production lines. These limitations provided the main motivation for our research, where we propose a machine learning approach based on acoustic signals that enables faster, cost-effective, and more practical defect detection for real-world applications.

11: At the end of the introduction, the authors should provide the reader with a clear summary of the novelty and methodology. This should include how the samples were prepared, how the algorithms were trained, and whether different algorithms were compared to find the best one. It should also explain how the accuracy of the models was evaluated, if train-test splitting was used to assess the ability to predict new samples, and what materials were tested. All these details should be clearly defined in the last paragraph of the introduction.

Answer: In the final paragraph of the introduction, we highlighted the research gap and the novelty of the study, while briefly outlining the main steps of the methodology, including sample preparation, algorithm training, and model evaluation. However, due to the page limit set by the journal, these methodological details were only mentioned concisely in the introduction and are discussed comprehensively in the Materials and Methods section

12: The methodology section of the paper is confusing and lacks a clear summary. I suggest the authors include a detailed flowchart that outlines the sequence of steps taken in the study. This would greatly improve the readability and help readers understand the research process more quickly.

Answer: We appreciate the reviewer’s valuable comment. In response, we have added a detailed flowchart (Figure 1) at the beginning of the Materials and Methods section to clearly illustrate the sequence of steps followed in this study. Furthermore, we have revised the opening paragraph of the section to provide a concise summary of the methodology before presenting each step in detail. These changes improve clarity and help readers better follow the research process.

13: When reporting a case study in a scientific paper, simply naming the very reliable vehicle (Tiba) equipment is not sufficient. You should provi

---

## [Decision Letter · Decision Letter 1]

9 Sep 2025

Dear Dr. Rohani,

Thank you for submitting your manuscript to PLOS ONE. After careful consideration, we feel that it has merit but does not fully meet PLOS ONE’s publication criteria as it currently stands. Therefore, we invite you to submit a revised version of the manuscript that addresses the points raised during the review process.

We look forward to receiving your revised manuscript.

Kind regards,

Mohammad Azadi

Academic Editor

PLOS ONE

Journal Requirements:

Additional Editor Comments:

Besides considering the reviewer's comments, there are still several issues on the revised manuscript, as follows,

1) There are two equations for the number of (1).

2) Almost all parts of the results section are yellow and it is confusing that either the discussion is improved or not.

3) The discussion is to described the details and reasons of the obtained behavior, plus the comparison of the results to others in references.

4) It is not clear which reference is new! No yellow one!

5) The conclusions is lengthy and it should be shortened.

Reviewers' comments:

Reviewer's Responses to Questions

**Comments to the Author**

Reviewer #1: (No Response)

Reviewer #2: All comments have been addressed

2. Is the manuscript technically sound, and do the data support the conclusions?

Reviewer #1: Partly

Reviewer #2: Yes

3. Has the statistical analysis been performed appropriately and rigorously?

Reviewer #1: (No Response)

Reviewer #2: N/A

4. Have the authors made all data underlying the findings in their manuscript fully available?

Reviewer #1: Yes

Reviewer #2: Yes

5. Is the manuscript presented in an intelligible fashion and written in standard English?

Reviewer #1: No

Reviewer #2: Yes

Reviewer #1: The revised files are not acceptable in their current form. The authors should clearly highlight only the exact changes made in the manuscript rather than highlighting entire sections. At present, large portions of the text are highlighted in yellow, even where no changes appear to have been made. This makes it very difficult to evaluate whether the authors have addressed my comments effectively.

I strongly request that:

1:The authors highlight only the specific text that was changed or added, not entire sections.

2:The response-to-reviewers document should clearly list and describe all changes made in the manuscript, including page and line numbers where applicable.

3:The highlighting should be revised to accurately reflect changes to ensure clarity during review.

With the current formatting, it is not possible to determine whether my previous comments have been adequately addressed.

Reviewer #2: The paper is now significantly improved and I am happy to recommend publication. Manuscript improved quality. It is acceptable.

**Do you want your identity to be public for this peer review?** For information about this choice, including consent withdrawal, please see our Privacy Policy

Reviewer #1: No

Reviewer #2: No

---

## [Author Response · Author response to Decision Letter 2]

10 Sep 2025

Manuscript Title:

Non-destructive defect detection in powder metallurgy components using acoustic signals and machine learning classification

Manuscript PONE-D-25-28300R2

Dear Editor in Chief,

We would like to sincerely thank you and the reviewers for the time and effort devoted to evaluating our manuscript. The constructive comments and insightful suggestions provided during the first round of review were invaluable in improving the overall quality and clarity of the work.

We also wish to express our gratitude to the reviewers for their valuable contributions. At the same time, we would like to apologize for any difficulties encountered in tracking the extensive revisions made in the first round. Due to the large volume of modifications, the traceability of changes may not have been sufficiently clear.

In this revised submission, we have carefully addressed all remaining concerns. To ensure clarity and ease of review, the modifications have been clearly highlighted and page numbers have been indicated: changes made in response to the Editor’s comments are highlighted in green, while those addressing the reviewers’ comments are highlighted in yellow.

We hope that the manuscript in its current form meets the journal’s standards and satisfactorily addresses all comments

Abbas Rohani

Additional Editor Comments:

Besides considering the reviewer's comments, there are still several issues on the revised manuscript, as follows,

1) There are two equations for the number of (1).

Answer: You are absolutely right, and the numbering of the formulas has been carefully reviewed and corrected. The revisions have been highlighted in green on page 17, within subsection “2.6. Evaluation of classifier performance”.

2) Almost all parts of the results section are yellow and it is confusing that either the discussion is improved or not.

Answer: As you are aware, during the first round of revision the volume of required modifications was substantial. Among the major changes, the introduction had to be condensed to a maximum of three pages based on your recommendation, and the overall English standard of the manuscript also needed significant improvement. In addition, several extensive revisions were requested across other sections of the paper. These combined efforts unfortunately led to difficulties in clearly tracking and distinguishing the revised parts, which may have caused confusion.

We sincerely apologize for this issue. In the current round, special care has been taken to precisely address and clearly highlight the changes within the text so that they can be easily traced by both you and the reviewers

3) The discussion is to described the details and reasons of the obtained behavior, plus the comparison of the results to others in references.

Answer: As was mentioned in the first round of revision, in accordance with your valuable suggestion and also your Comment No. 6 from the initial review, Table 10 has been added on page 31 together with its explanation. In addition, the discussion has been further expanded and the results have been compared with those of other studies on pages 29–30. These modifications, which were made to strengthen the discussion and comparison of findings, have been highlighted in green for clarity. We hope that these revisions satisfactorily address your concern

4) It is not clear which reference is new! No yellow one!

Answer: We sincerely thank you for your kind remark. You are absolutely right, and this point was unfortunately overlooked in the first round of revision. In the current version, the newly added references have been clearly highlighted in green. Please kindly note that references 31 to 53 are new additions to the reference list. These can be found on the last three pages of the manuscript (pp. 36–38).

5) The conclusions is lengthy and it should be shortened.

Answer: As noted in Comment No. 18 from Reviewer 1, the conclusion section was previously revised to present the main findings in bullet points, including both qualitative and quantitative results. However, in accordance with your valuable recommendation, this section has now been further shortened while still retaining the key outcomes of the study. We hope that the revised version addresses your concern appropriately (pp. 32–33).

Reviewer # 1

The revised files are not acceptable in their current form. The authors should clearly highlight only the exact changes made in the manuscript rather than highlighting entire sections. At present, large portions of the text are highlighted in yellow, even where no changes appear to have been made. This makes it very difficult to evaluate whether the authors have addressed my comments effectively.

I strongly request that:

1:The authors highlight only the specific text that was changed or added, not entire sections.

2:The response-to-reviewers document should clearly list and describe all changes made in the manuscript, including page and line numbers where applicable.

3:The highlighting should be revised to accurately reflect changes to ensure clarity during review.

With the current formatting, it is not possible to determine whether my previous comments have been adequately addressed.

Answer: We fully agree with your observation and sincerely apologize for the difficulties caused. In the first round of revision, the extensive changes requested by the Editor—including condensing the introduction to a maximum of three pages, revising the conclusions and other sections, and improving the manuscript’s English—resulted in large portions of the text being highlighted in yellow.

We acknowledge that this made it difficult to track the specific changes corresponding to your comments. In the current revision, only the modifications related to your comments are highlighted in yellow, while changes requested by the Editor are highlighted in green. We also note that the changes suggested by Reviewer #2, which were already addressed in the first round, were not highlighted previously.

Finally, all of your comments have been carefully addressed again in the response-to-reviewers document, with precise page and line references provided to ensure clarity and traceability of each modification.

We hope that these revisions now meet your expectations

I recommend the paper for major revision with the following comments:

1: The title of the paper is too general, especially since the authors mention “powder metallurgy components” without specifying the material. This approach overgeneralizes the results obtained from a limited number of samples to the entire manufacturing process.

Answer: Thank you for your valuable comment. We agree with your observation that the previous title was too general. To address this concern and to more accurately reflect the specific scope of our study, we have revised the title to:

“Non-destructive defect detection in powder metallurgy automotive oil pump stators using acoustic signals and machine learning classification.”

This revision avoids overgeneralization and specifies the exact type of component investigated, ensuring that the results are clearly contextualized within automotive oil pump stators manufactured by powder metallurgy.

The change has been made on page 1 of the manuscript.

2: You should avoid using overly polished or complicated text in a research paper. For example, the sentence “Furthermore, PM manufacturing typically involves high production volumes, necessitating not only accurate defect detection but also rapid inspection methods” is unnecessarily complex. It uses correlative conjunctions within a present participle clause, which makes it difficult to follow. Also, avoid combining phrases like “due to” and “however” in one sentence. It would be clearer to restructure the sentence using a single transition word such as “although.” Initially, I mentioned this issue in the abstract, but after reviewing the rest of the paper, I recommend checking the entire text. Although the language is grammatically correct, it is sometimes hard to read. A research paper should use clear, direct, and consistent structures — it should not include a variety of sentence forms like TOEFL or IELTS essays.

Answer: We have carefully reviewed the entire manuscript and revised the text wherever possible to simplify complex sentences, improve clarity, and ensure direct and consistent expression throughout the paper. In particular, the introduction was rewritten and condensed according to the Editor’s recommendation to limit it to three pages, which involved removing or summarizing many sentences. Moreover, the entire manuscript was reviewed again based on your suggestion, and numerous improvements were made to enhance readability and clarity. We hope these revisions address your concerns satisfactorily.

Specifically, the sentence you highlighted in the abstract has been rewritten and modified (see page 1, line 5), and we note that the remainder of the abstract has also been revised in accordance with the recommendations of the Editor and Reviewer #2.

3: The advantage of using machine learning (ML) in this research is not mentioned in the abstract. It should be explained how this method can be effective and how it compares with other conventional methods. Currently, the abstract only highlights the necessity of using acoustic signal testing, without emphasizing the role or benefits of ML. Additionally, instead of providing a long general introduction in the abstract, it would be more useful to include details about the algorithm training process. Overall, the abstract needs to be completely restructured.

Answer: We sincerely thank the Reviewer for this valuable comment. In the revised version, the Abstract has been improved to explicitly highlight the advantages of machine learning over conventional threshold-based inspection techniques, emphasizing its ability to detect complex, non-linear acoustic signal patterns that are often missed by traditional approaches. In addition, the training process of the algorithms has been clarified by specifying that time- and frequency-domain features were extracted from the collected acoustic data and used to train and validate four supervised ML classifiers (SVM, k-NN, MLP, and RBF). Furthermore, the comparative results of these models have been briefly summarized to demonstrate their performance, with the RBF model achieving 100% accuracy.

The changes can be found in the Abstract on page 1 of the manuscript.

4: How is it possible that SVM and MLP neural network are listed as keywords in your paper but are not mentioned in the title or abstract? These important terms should at least appear in the abstract to reflect the main methods used in the study.

Answer: We appreciate the Reviewer’s insightful observation. In accordance with your suggestion and the Editor’s guidance, the list of keywords has been carefully revised, and the Abstract has been rewritten to ensure that all major methods, including SVM and MLP, are explicitly mentioned. This issue has been fully addressed in the revised manuscript, and the updated list of keywords can be reviewed for confirmation

The changes can be found in the Keywords section on page 1 of the manuscript.

5: Please avoid introducing abbreviations for the first time without proper definition. When using an abbreviation, it should be introduced once with its full form, and then only the abbreviation should be used throughout the paper. For example, you have introduced “PM” and “ML” several times in different sections. Please check and correct this issue for all abbreviations in the text.

Answer: We sincerely thank the Reviewer for this valuable comment. Following your suggestion, we have carefully revised the manuscript to ensure that each abbreviation is introduced only once with its full form at the first appearance, and then consistently used throughout the text. Specifically, abbreviations such as “PM” and “ML” have been corrected accordingly. This revision has been applied uniformly across all sections of the manuscript.

The corrections are reflected throughout the manuscript.

6: In the introduction, several long sentences make it difficult to follow the ideas clearly. The sentence structures are similar to those typically found in TOEFL reading passages, which are unnecessarily complex for a research paper. I recommend simplifying and restructuring these sentences to improve clarity and readability.

Answer: We appreciate the reviewer’s valuable comment. Following the editor’s guidance to reduce the introduction to a maximum of three pages, this section has been revised accordingly. In addition, the issue rightly noted by the reviewer regarding long and complex sentence structures has been carefully addressed, not only in the introduction but also throughout other sections of the manuscript to enhance clarity and readability.

Since the entire introduction was extensively rewritten, all the points you highlighted have been incorporated. Please refer to pages 3–5 of the manuscript to review these changes.

7: The introduction is too long, and the ideas in the paragraphs are not well connected. As a result, it is difficult for the reader to follow the overall direction of the paper. Moreover, the extended explanation about different types of non-destructive inspection methods makes the introduction unnecessarily lengthy and somewhat boring. In particular, the second paragraph does not contribute to the main objective of the paper. Instead, it seems to list information without offering a clear link to the research focus. Therefore, I suggest reorganizing and shortening the introduction to improve clarity, relevance, and reader engagement.

Answer: We sincerely appreciate your valuable feedback. In line with your recommendation and the editor’s suggestion, the introduction has been thoroughly revised and reorganized. The section was shortened to improve clarity, eliminate redundancies, and strengthen the logical flow of ideas. The part concerning non-destructive inspection methods was also streamlined to ensure relevance to the research objectives. We hope that these revisions adequately address your concern.

Based on both your comment and the Editor’s guidance, the introduction was reduced to three pages and substantially rewritten; therefore, we did not highlight the entire section. However, the last two paragraphs of the introduction were specifically revised in response to this comment and highlighted in yellow for your convenience (pages 5). Please refer to pages 3–5 of the manuscript to review these changes.

8: The long explanation of vibration non-destructive testing in paragraph 3 of the introduction may be boring for the reader. It would be better to shorten this section and focus more on information directly relevant to the study.

Answer: We appreciate your observation. This section has been shortened and simplified to avoid unnecessary detail. The primary aim of including it was to introduce the subject and highlight the importance of applying machine learning methods in this research area. In line with your valuable feedback, we have revised and restructured this part to improve clarity and relevance.

9: The literature review in the introduction related to acoustic testing is insufficient. Instead of listing different types of quality control methods and inspections in paragraph 2 and providing a long introduction to vibration testing at the beginning of paragraph 3, please include more detailed information about how acoustic testing differs from other methods. Also, explain the necessity of using this method compared to others, recent developments in acoustic testing, and its advantages.

Answer: We appreciate your comment regarding the coverage of acoustic testing in the introduction. The introduction has been thoroughly revised and streamlined to avoid unnecessary elaboration. However, based on our review and the available references, there is very limited research on defect detection in industrial components produced by powder metallurgy, particularly in the automotive sector. Therefore, to highlight the importance of acoustic testing, we referred to related studies where applicable, in order to clarify its relevance and advantages compared to other methods. Additionally, Table 10

---

## [Decision Letter · Decision Letter 2]

15 Sep 2025

Dear Dr. Rohani,

Thank you for submitting your manuscript to PLOS ONE. After careful consideration, we feel that it has merit but does not fully meet PLOS ONE’s publication criteria as it currently stands. Therefore, we invite you to submit a revised version of the manuscript that addresses the points raised during the review process.

We look forward to receiving your revised manuscript.

Kind regards,

Mohammad Azadi

Academic Editor

PLOS ONE

Journal Requirements:

Additional Editor Comments:

The conclusions section is still lengthy.

Reviewers' comments:

Reviewer's Responses to Questions

**Comments to the Author**

Reviewer #1: (No Response)

2. Is the manuscript technically sound, and do the data support the conclusions?

Reviewer #1: Partly

3. Has the statistical analysis been performed appropriately and rigorously?

Reviewer #1: I Don't Know

4. Have the authors made all data underlying the findings in their manuscript fully available?

Reviewer #1: Yes

5. Is the manuscript presented in an intelligible fashion and written in standard English?

Reviewer #1: Yes

Reviewer #1: Dear Editor,

I cannot reject this paper, but I also cannot accept it.Therefore, I recommend that additional reviewers carefully evaluate the manuscript in this major revision, but I must emphasize that this will be the final review round. I will not accept further revisions beyond this because I am doubtful about the accuracy of this paper. While the methodology and overall work are promising, the authors have been inaccurate in addressing even minor comments. This raises concerns about the reliability of the presented results and representations. I am doubtful that other, more significant issues may exist that I have not identified.

Several examples of minor issues include:

Comment 17: The authors responded that “Table 7 has been removed and replaced with Figure 8.” However, Figure 8 does not exist in the revised manuscript.

Abbreviations: Inconsistencies remain for terms such as “machine learning” (sometimes written in full, sometimes abbreviated), “accuracy” (defined as Acc but inconsistently used), and NN. Despite the authors’ claim that abbreviations were corrected, the manuscript still shows these inconsistencies.

Highlighting changes: I requested that revisions be highlighted. This was not done. For example, in the abstract, a sentence about acoustic signals from different components was revised but not highlighted.

Comment 16: My suggestion was to include the formulation of kernel functions, which is standard in scientific papers. Instead, the authors focused on a comparison between supervised and unsupervised machine learning, which was not requested.

**Do you want your identity to be public for this peer review?** For information about this choice, including consent withdrawal, please see our Privacy Policy

Reviewer #1: No

---

## [Author Response · Author response to Decision Letter 3]

20 Sep 2025

Manuscript Title:

Non-destructive defect detection in powder metallurgy components using acoustic signals and machine learning classification

Manuscript PONE-D-25-28300R3

Dear Editor in Chief,

We sincerely thank you and the reviewers for the time and effort devoted to evaluating our manuscript. The constructive comments and insightful suggestions provided during the review process were invaluable in improving the quality, clarity, and overall presentation of our work.

We also wish to express our gratitude to the reviewers for their detailed and professional feedback. At the same time, we apologize for any difficulties encountered in tracking the extensive revisions made in the second round. Due to the substantial number of modifications, some changes may not have been immediately evident, and we appreciate your understanding in this regard.

We believe that the manuscript, in its current revised form, fully addresses the reviewers’ comments and meets the standards of PLOS ONE. We hope it will now be considered suitable for publication.

Additional Editor Comments:

1) The conclusions section is still lengthy.

Answer: The Conclusions section has been further revised and shortened to ensure conciseness as requested.

" Conclusions

This study demonstrated that acoustic resonance testing combined with ML classifiers can accurately detect defects in powder metallurgy components. The main findings are summarized as follows:

SVM achieved 99% accuracy, KNN reached 100% accuracy, and both MLP and RBF classifiers showed near-perfect training and testing performance. These results demonstrate the robustness and reliability of the models.

KNN and RBF achieved perfect overall performance, while SVM and MLP reached approximately 0.99 accuracy with minor false positives. RBF maintained near-perfect generalization using only 50% of the training data. In contrast, SVM and MLP were more sensitive to smaller datasets, and KNN showed slight decreases in testing accuracy.

Carefully chosen RBF features from the third and fourth frequency ranges enabled perfect classification, including four-class distinction. This shows that dimensionality can be reduced without losing robustness and highlights the importance of mid-to-high frequency features for distinguishing defect types.

Acoustic resonance testing combined with ML achieved 100% accuracy, outperforming other non-destructive methods. This approach enables reliable quality control and fault diagnosis of PM components using 10–20 kHz frequency bands.

In conclusion, combining acoustic resonance testing with ML provides a robust, efficient, and scalable approach for inspecting PM components. Although the method is highly accurate, it does not provide information on defect location or severity, which limits its applicability in industrial decision-making. Future work should focus on developing methods for defect localization and severity estimation, implementing automated feature selection, enabling real-time analysis, and extending the approach to other industrial components."

Reviewer # 1

I cannot reject this paper, but I also cannot accept it. Therefore, I recommend that additional reviewers carefully evaluate the manuscript in this major revision, but I must emphasize that this will be the final review round. I will not accept further revisions beyond this because I am doubtful about the accuracy of this paper. While the methodology and overall work are promising, the authors have been inaccurate in addressing even minor comments. This raises concerns about the reliability of the presented results and representations. I am doubtful that other, more significant issues may exist that I have not identified.

Answer: We sincerely thank the reviewer for the insightful and constructive feedback, which has substantially improved the quality and clarity of this manuscript.

Because the first revision required extensive rewriting, many sections were fully restructured.

In the second round, highlights were limited to the unresolved points to avoid confusion, which may have made some changes less visible.

We apologize for any inconvenience and summarize below the key revisions implemented.

Overview of major revisions implemented in response to the constructive comments and recommendations provided during both the first and second rounds of review.

Title Refinement – Revised to “Non-destructive defect detection in powder metallurgy automotive oil pump stators using acoustic signals and machine learning classification” to ensure specificity and avoid overgeneralization.

Language and Style – Entire manuscript carefully edited to shorten sentences, improve readability, and maintain a clear, direct scientific tone.

Abstract Enhancement – Completely rewritten to emphasize the advantages of machine learning, include key training details, and report the performance of the four classifiers (SVM, k-NN, MLP, RBF). Keywords updated accordingly.

Introduction – Reduced to three pages, reorganized for logical flow, and revised to clearly present the research gap, motivation for ML, and novelty while removing redundant NDT descriptions.

Methodology – Added a detailed flowchart (new Figure 1) summarizing experimental and modeling steps; opening paragraph rewritten for clarity.

Case Study and Defects – Provided full context on the Tiba vehicle, production site, defect types, and sample collection procedures.

Modeling Details – Expanded explanation of KNN parameters, kernel settings, and validation strategy (5-fold cross-validation with 20 runs); tables and figures updated with mean values and standard deviations.

Results Presentation – Replaced Table 7 with a new Figure 6 showing training-set size effects on accuracy with standard deviations.

Conclusions – Shortened and reformatted into concise bullet points summarizing key quantitative and qualitative findings, industrial implications, and directions for future research.

Several examples of minor issues include:

1. Comment 17: The authors responded that “Table 7 has been removed and replaced with Figure 8.” However, Figure 8 does not exist in the revised manuscript.

Answer: We appreciate the reviewer’s careful examination. It appears there was a mislabeling in our previous response. In the revised manuscript, Figure 6, not Figure 8, has replaced Table 7. Please see the top of page 25, where Figure 6 presents the corresponding results. We apologize for the confusion and thank the reviewer for noting this oversight.

Figure 6. Impact of training set size on the mean and standard deviation of SVM, KNN, MLP, and RBF classification accuracy under cross-validation

2. Abbreviations: Inconsistencies remain for terms such as “machine learning” (sometimes written in full, sometimes abbreviated), “accuracy” (defined as Acc but inconsistently used), and NN. Despite the authors’ claim that abbreviations were corrected, the manuscript still shows these inconsistencies.

Answer: We sincerely thank the reviewer for pointing out this important issue. The entire manuscript has been thoroughly re-checked, and all abbreviations, including machine learning (ML), accuracy (Acc), and NN, have now been consistently revised throughout the text to ensure uniform usage.

Machine learning (ML): Introduced in full at first mention and subsequently abbreviated as ML on pages 1, 3, 4, 5, 6, 7, 10, 29, 30, 31, 32, and 33.

Accuracy (Acc): Used in full throughout the text, except in Table 6 (page 23), where it is presented as Acc together with other classifier evaluation metrics (e.g., TP, FP, Acc, Prec, Rec, F1, Spec, AUC, YI) due to space limitations. A note beneath the table clearly defines the abbreviation to maintain clarity.

NN: The inconsistency in Table 10 has been corrected to ensure proper and consistent usage.

3. Highlighting changes: I requested that revisions be highlighted. This was not done. For example, in the abstract, a sentence about acoustic signals from different components was revised but not highlighted.

Answer: We sincerely apologize for the inconvenience caused. As requested by the editor and reviewers, the entire abstract was extensively rewritten to incorporate all comments and suggestions. Because the revisions involved a complete rephrasing rather than isolated sentence changes, it was not possible to highlight each specific modification individually, which may have made it difficult to track the changes. We kindly ask the reviewer to review the abstract again with this in mind, and we once again apologize for any difficulty this may have created.

4. Comment 16: My suggestion was to include the formulation of kernel functions, which is standard in scientific papers. Instead, the authors focused on a comparison between supervised and unsupervised machine learning, which was not requested.

Answer: We sincerely thank the reviewer for the valuable suggestion. In accordance with your recommendation, the following content has been added to the Materials and Methods section (pages 14–15) to include the formulations of the kernel functions:

" To handle nonlinear relationships, four kernel functions were evaluated: linear, polynomial of degree 2, polynomial of degree 3, and the RBF. These kernels allow the input features to be mapped into higher-dimensional spaces, improving class separability. Their mathematical formulations are presented in Eqs. (2)–(5):

K(x_i,x_j )=x_i^T x_j (2)

K(x_i,x_j )=〖(〖γx〗_i^T x_j+r)〗^2 (3)

K(x_i,x_j )=〖(〖γx〗_i^T x_j+r)〗^3 (4)

K(x_i,x_j )=exp⁡(-γ‖x_i-x_j ‖^2) (5)

where xi and xj represent input feature vectors, γ>0 denotes the kernel scale parameter that controls the influence of individual samples, and rrr is the coefficient used in the polynomial kernels. These kernel functions enable the SVM classifier to construct both linear and highly nonlinear decision boundaries, thereby improving classification performance in high-dimensional datasets."

Reviewer # 2

The paper is now significantly improved and I am happy to recommend publication. Manuscript improved quality. It is acceptable.

Answer: We sincerely appreciate your valuable comments and constructive feedback, which have greatly contributed to improving the quality of our manuscript. We also apologize for any inconvenience caused by the insufficient traceability of the revisions in the earlier round. Your support and recommendation for publication are highly appreciated.

We sincerely thank you for your insightful and constructive comments. Your valuable feedback has been instrumental in refining and improving the quality of our manuscript. We remain fully open to any additional suggestions you may have and are committed to incorporating all relevant recommendations in the final version of the paper.

Abbas Rohani

Department of Biosystems Engineering, Faculty of Agriculture, Ferdowsi University of Mashhad, Mashhad, Iran

ORCID: https://orcid.org/0000-0002-4494-7058

Corresponding author: Abbas Rohani, Email: arohani@um.ac.ir

---

## [Decision Letter · Decision Letter 3]

28 Sep 2025

Non-destructive defect detection in powder metallurgy automotive oil pump stators using acoustic signals and machine learning classification

PONE-D-25-28300R3

Dear Dr. Rohani,

We’re pleased to inform you that your manuscript has been judged scientifically suitable for publication and will be formally accepted for publication once it meets all outstanding technical requirements.

Kind regards,

Mohammad Azadi

Academic Editor

PLOS ONE

Additional Editor Comments (optional):

Reviewers' comments:

Reviewer's Responses to Questions

**Comments to the Author**

Reviewer #1: All comments have been addressed

Reviewer #3: (No Response)

2. Is the manuscript technically sound, and do the data support the conclusions?

Reviewer #1: Partly

Reviewer #3: Yes

3. Has the statistical analysis been performed appropriately and rigorously?

Reviewer #1: I Don't Know

Reviewer #3: Yes

4. Have the authors made all data underlying the findings in their manuscript fully available?

Reviewer #1: Yes

Reviewer #3: Yes

5. Is the manuscript presented in an intelligible fashion and written in standard English?

Reviewer #1: Yes

Reviewer #3: Yes

Reviewer #1: Dear Editor, I accept the paper. The machine learning training is well-executed, but the mechanical-based results, while similar to the ML outcomes, may contain inaccuracies. I recommend further evaluation by an expert in solid mechanics.

Reviewer #3: I have reviewed the comments and the paper. While the concern of Reviewer 1 regarding the results and their persistence in the review process is acknowledged, I can confirm that the paper is well-organized and suitable for publication after three rounds of revision

**Do you want your identity to be public for this peer review?** For information about this choice, including consent withdrawal, please see our Privacy Policy

Reviewer #1: No

Reviewer #3: **Yes**

---

## [Editor Report · Acceptance letter]

PONE-D-25-28300R3

PLOS ONE

Dear Dr. Rohani,

I'm pleased to inform you that your manuscript has been deemed suitable for publication in PLOS ONE. Congratulations! Your manuscript is now being handed over to our production team.

Kind regards,

on behalf of

Dr. Mohammad Azadi

Academic Editor

PLOS ONE